# Interferon priming is essential for human CD34+ cell-derived plasmacytoid dendritic cell maturation and function

A. Laustsen[1], R.O. Bak [1,2,3], C. Krapp[1], L. Kjær[1], J.H. Egedahl[1,4,5], C.C. Petersen[1], S. Pillai[6], H.Q. Tang[7], N. Uldbjerg[7], M. Porteus[3], N.R. Roan[4,5], M. Nyegaard [1], P.W. Denton [8,9] & M.R. Jakobsen [1]

Plasmacytoid dendritic cells (pDC) are essential for immune competence. Here we show that pDC precursor differentiated from human CD34+ hematopoietic stem and progenitor cells (HSPC) has low surface expression of pDC markers, and has limited induction of type I interferon (IFN) and IL-6 upon TLR7 and TLR9 agonists treatment; by contrast, cGAS or RIG-I agonists-mediated activation is not altered. Importantly, after priming with type I and II IFN, these precursor pDCs attain a phenotype and functional activity similar to that of peripheral blood-derived pDCs. Data from CRISPR/Cas9-mediated genome editing of HSPCs further show that HSPC-pDCs with genetic modifications can be obtained, and that expression of the IFN-α receptor is essential for the optimal function, but dispensable for the differentiation, of HSPC-pDC percursor. Our results thus demonstrate the biological effects of IFNs for regulating pDC function, and provide the means of generating of gene-modified human pDCs.

[1] Department of Biomedicine, Aarhus University, Wilhelm Meyers Alle 4, 8000 Aarhus C, Denmark. [2] Aarhus Institute of Advanced Studies (AIAS), Aarhus University, Høegh-Guldbergs Gade 6B, 8000 Aarhus C, Denmark. [3] Department of Pediatrics, Stanford University, Stanford, CA 94305, USA. [4] Department of Urology, University of California, San Francisco, CA 94158, USA. [5] The J. David Gladstone Institutes, San Francisco, CA 94158, USA. [6] University of California, San Francisco, Blood Systems Research Institute, 270 Masonic Avenue, San Francisco 94118-4417 CA, USA. [7] Department of Obstetrics and Gynaecology, Aarhus University Hospital Skejby, Aarhus 8200, Denmark. [8] Department of Infectious Diseases, Aarhus University Hospital Skejby, Aarhus 8200, Denmark. [9] Department of Clinical Medicine, Aarhus University Hospital Skejby, Aarhus 8200, Denmark. These authors contributed equally: R.O. Bak, C. Krapp. Correspondence and requests for materials should be addressed to M.R.J. (email: mrj@biomed.au.dk)

Plasmacytoid dendritic cells (pDCs) are key effectors in cellular immunity with the ability to not only initiate immune responses but also to induce tolerance to exogenous and endogenous antigens[1]. pDCs are distinct from conventional DCs (cDCs) as they express high levels of interferon regulatory factor 7 (IRF7) and they primarily sense pathogens through Toll-like receptor (TLR) 7 and 9[1,2]. Through these pattern-recognition receptors, pathogen-derived nucleic acids can activate pDCs to produce high levels of type I interferon (IFN). Furthermore, activated pDCs link the innate and adaptive immune system together via cytokine production combined with antigen-presenting cell (APC) activity. This dual role makes pDCs essential for attaining an antiviral state during infections, propagating adjuvant activity in the context of vaccination, and promoting immunogenic antitumor responses[1,3,4]. Conversely, a delicate balance is essential, as hyper-activation of pDCs has been associated with the pathogenesis of several diseases including viral infections, systemic lupus erythematosus (SLE), and tumorigenesis[1,2]. Collectively, pDCs have a key and multifaceted role in the immune system, prompting research into their development and mechanisms of action.

IFNs constitute pleiotropic effectors, which exert a fundamental role during inflammation and immune responses, and recently their activity has been implicated in regulating hematopoietic homeostasis[5–7]. Mice deficient for the type I IFN receptor (IFNAR) and STAT2 show decreased numbers of pDCs within peripheral blood, as well as reduced capacity to mature upon TLR7 stimulation[8,9]. Moreover, neonatal mice stimulated with type I IFN were found to have increased numbers of pDCs in the periphery[8]. Accordingly, type I IFNs have been shown to promote the expression of Flt3, the receptor for Flt3-L, which is known to be one of the primary physiological factors mediating DC homeostasis[10–12]. On the other hand, type I IFNs that were induced during systemic viral infections were found to negatively regulate pDC numbers via the intrinsic apoptosis pathway[13]. Thus, IFNs appear to regulate key aspects of murine pDC biology, including development, apoptosis, migration, homeostasis, and function. However, their role in dictating human pDC development and functions remains incompletely understood. Lack of biological studies of human pDCs can be ascribed to the very low frequencies of pDCs within peripheral blood, as pDCs represent less than 0.1% of peripheral blood mononuclear cells (PBMCs) and have a short lifespan ex vivo of 1–3 days[14]. Moreover, genetic manipulation of immune cells, in particular pDCs, remains a major challenge as most attempts of introducing genetic material (e.g., siRNAs) elicit an innate immune response[15,16]. While studies show that this effect can be reduced with the incorporation of 2′-O-methyl modifications and using specific methods of delivery, precise CRISPR/Cas9 gene editing in pDCs has yet to be reported, and the poor viability of PBMC-derived pDCs in cell culture remains a key issue, as the time to induce, observe, and analyze a corresponding phenotype is highly limited[15–19]. Thus, technologies that can genetically alter precursor cells prior to the development of pDCs would be a powerful method to circumvent these limitations.

Previous attempts to investigate pDC biology in vitro have led to the discovery of numerous culture conditions that support both expansion and differentiation of CD34[+] HSPCs into cells with pDC-like phenotypic characteristics[20–25]. However, the total yield of pDCs has been low, and the similarity to blood-derived pDCs has thus far not been validated. Importantly, the biological effects of IFNs on human pDC maturation and function has not been investigated.

Here we describe a culture protocol focusing on the yield and biological characteristics of human pDCs. We show that by using an optimized combination of cytokines and growth factors, an average of 81 (±20) pDCs per single HSPC can be generated, which outperforms other available protocols. Upon extensive evaluation of the functional and phenotypic features of the HSPC-derived pDCs in comparison to canonical peripheral blood-derived pDCs, we identify a hitherto unknown precursor pDC phenotype that displays low expression of pDC-specific markers and poor functional activation when stimulated with synthetic TLR7 and TLR9 agonists. Upon priming with type I and II IFNs, we find that this immature phenotype shifts toward a canonical pDC phenotype with full functional activity, highlighting the requirement of positive stimuli in the form of IFNs during in vitro human pDC development. Notably, we also demonstrate that genetically modified pDCs can be derived from CRISPR/Cas9 gene-edited CD34[+] HSPCs, and that knockout of the type I IFN receptor abrogates TLR7 and TLR9-mediated innate sensing.

## Results

**Expansion and differentiation of HSPCs pDC**. Initially, we investigated various culture conditions reported earlier to support HSPC survival, proliferation, and differentiation toward cells with pDC lineage characteristics. In validation of our experimental approach, we first reproduced outcomes from past published protocols and made direct comparisons with our culture system[20–25] (Fig. 1a, b). As baseline condition, we implemented a previously reported pDC differentiation protocol entailing culturing cells 21 days in medium containing the cytokines and growth factors Flt3-L, TPO, and interleukin-3 (IL-3)[20]. To stimulate expansion of progenitor cells during differentiation we added SCF[26] and SR1[27], and observed far greater expansion when both factors were included (240 fold ± 42 compared to 35 fold ± 26 in baseline culture; Fig. 1b). Blood-derived pDCs lack the expression of lineage-specific surface markers (i.e. CD3, CD14, CD16, CD19, CD20, and CD56) and the conventional DC marker CD11c[1]. Therefore, after the 21-day in vitro differentiation protocol, we enriched for pDCs by negative selection for lin[neg]CD11c[neg] cells by immunomagnetic separation and quantified the cells obtained. For simplicity these cells are hereafter referred to as HSPC-pDCs, despite the fact that no further phenotypic validation had been performed (Fig. 1c, d and Supplementary Fig. 1a–c). Importantly, SCF and SR1 proved to be indispensable additives to achieve an abundant and pure population of HSPC-pDC. We obtained an average yield of 81 (±20) HSPC-pDCs per single HSPC, corresponding to a total of $16.2 \times 10^6$ HSPC-pDCs ± $4.2 \times 10^6$ per starting $0.2 \times 10^6$ HSPCs. This protocol is a 57-fold improvement in pDC yield relative to the published protocol of generating HSPC-derived pDCs[20].

**HSPC-pDCs display an immature precursor phenotype**. Next, we examined whether the generated HSPC-pDCs were phenotypically and functionally equivalent to blood pDCs. Blood pDCs are characterized as being lin[neg]CD11c[neg], while being positive for CD123 (IL3Rα), CD303 (BDCA2), CD304 (BDCA4), CD4, and HLA-DR[1,28–30]. At culture day 21, we observed that the majority of HSPC-pDCs were positive for CD123, CD303, CD4, and HLA-DR, but only about 10% expressed CD304 (Fig. 2a and Supplementary Fig. 2a, b). Compared to blood-collected pDCs, HSPC-pDCs had significantly lower surface expression levels of these receptors as determined by mean fluorescence intensity (MFI) values (Fig. 2b and Supplementary Fig. 2b, c). While the HSPC culture conditions utilized were initially essential for expansion and survival, we hypothesized that continued incubation of cells in this cocktail arrested development of HSPC-pDCs. Such an outcome could result in the observed deficiencies relative

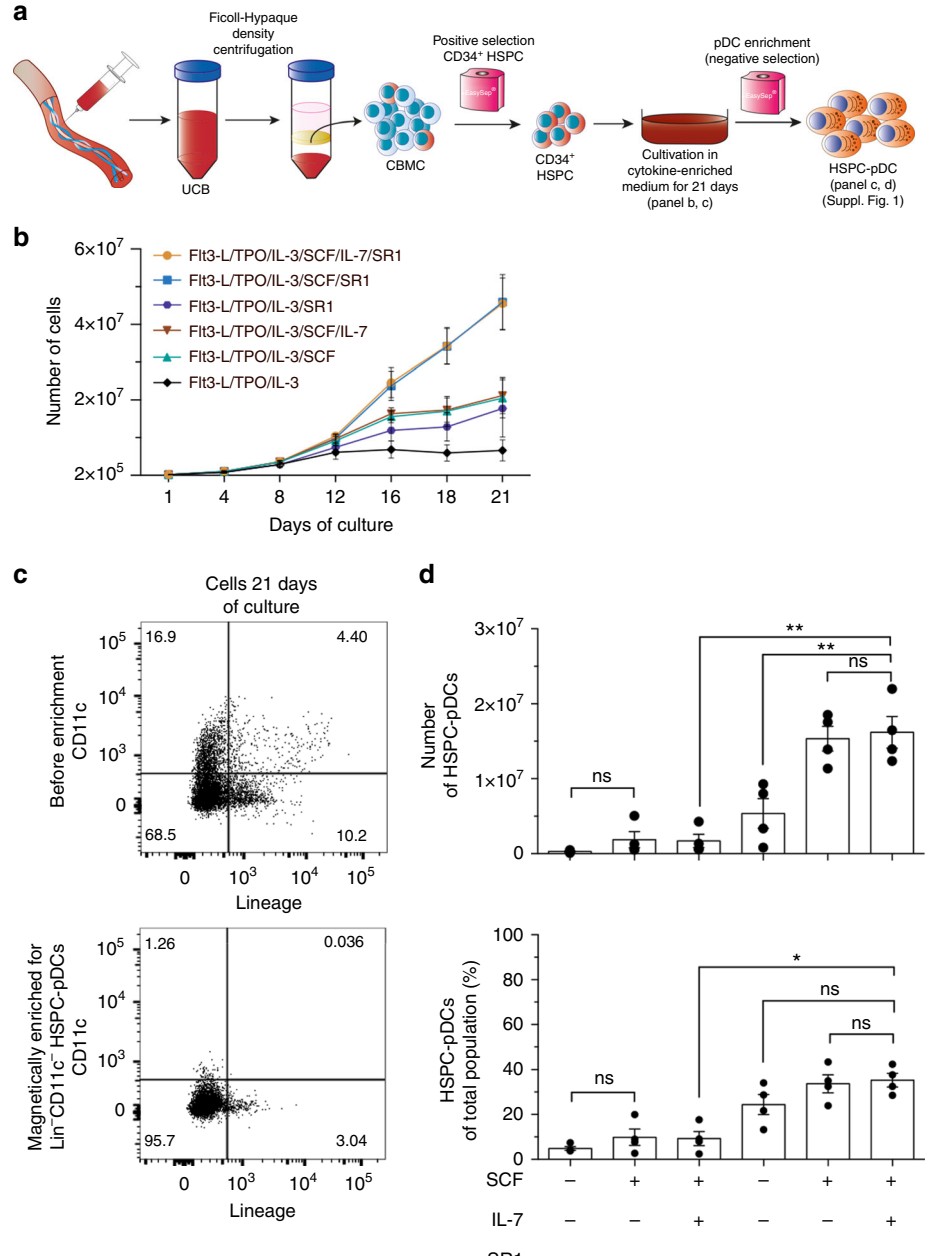

**Fig. 1** Generation of pDCs from hematopoietic stem and progenitor cells (HSPC-pDC). **a** Graphical illustration of the generation of HSPC-pDCs from HSPCs. CD34$^+$ HSPC were isolated from human umbilical cord blood (CB). Briefly, cord blood mononuclear cells (CBMC) were recovered by Ficoll-Hypaque density-gradient centrifugation. CD34$^+$ HSPCs were then isolated using anti-CD34 immunomagnetic beads. Next, $2 \times 10^5$ CD34$^+$ HSPC were cultivated for 21 days with factors promoting the expansion and differentiation of cells. At day 21, the cell suspension was magnetically enriched for lin$^{neg}$CD11c$^{neg}$ pDCs using negative immunomagnetic selection. **b** Total numbers of suspension cells measured at the experiment start (day 1) and the consecutive culture days (4, 8, 12, 16, 18, and 21) with the indicated factors in the medium. **c** Representative flow cytometry plots showing lineage and CD11c markers before and after enrichment of pDCs. **d** Total numbers (upper panel) or percent yield (lower panel) of HSPC-pDCs generated after magnetic enrichment (negative selection) at day 21 from Flt3-L, TPO, IL-3 culture supplemented with indicated growth factors. Error bars represent ± SEM of four donors. Statistical analysis was performed using one-way ANOVA (**d**) followed by Bonferroni post hoc test

to blood pDCs. To test this, HSPC-pDCs were cultured for an additional 24 h (Day 21 + 1) in medium supplemented with only IL-3 to support survival (Fig. 2c, d). Removal of growth factors led to an increase in surface expression of all the pDC-related markers compared to cells on day 21, but the emerging phenotype still did not recapitulate the phenotype of blood pDCs (Fig. 2d and Supplementary Fig. 3a–c). We then examined whether these HSPC-pDCs at day 21 or day 21 + 1 were able to respond to classical pDC activators, i.e. synthetic TLR7 and TLR9 agonists.

Both cell populations were found to produce significantly less type I interferon and IL-6 compared to blood pDCs (Fig. 2e, f). Nevertheless, upon stimulation with lipofectamin-formulated agonists directed at cytosolic DNA and RNA sensors, HSPC-pDCs produced high levels of type I IFN, indicating that HSPC-pDCs do not lack the capacity to induce innate immune responses (Fig. 2g). However, as was the case for TLR stimulation, the type I IFN responses was found to be lower compared to blood pDCs. Taken together, we hypothesize that day 21 HSPC-

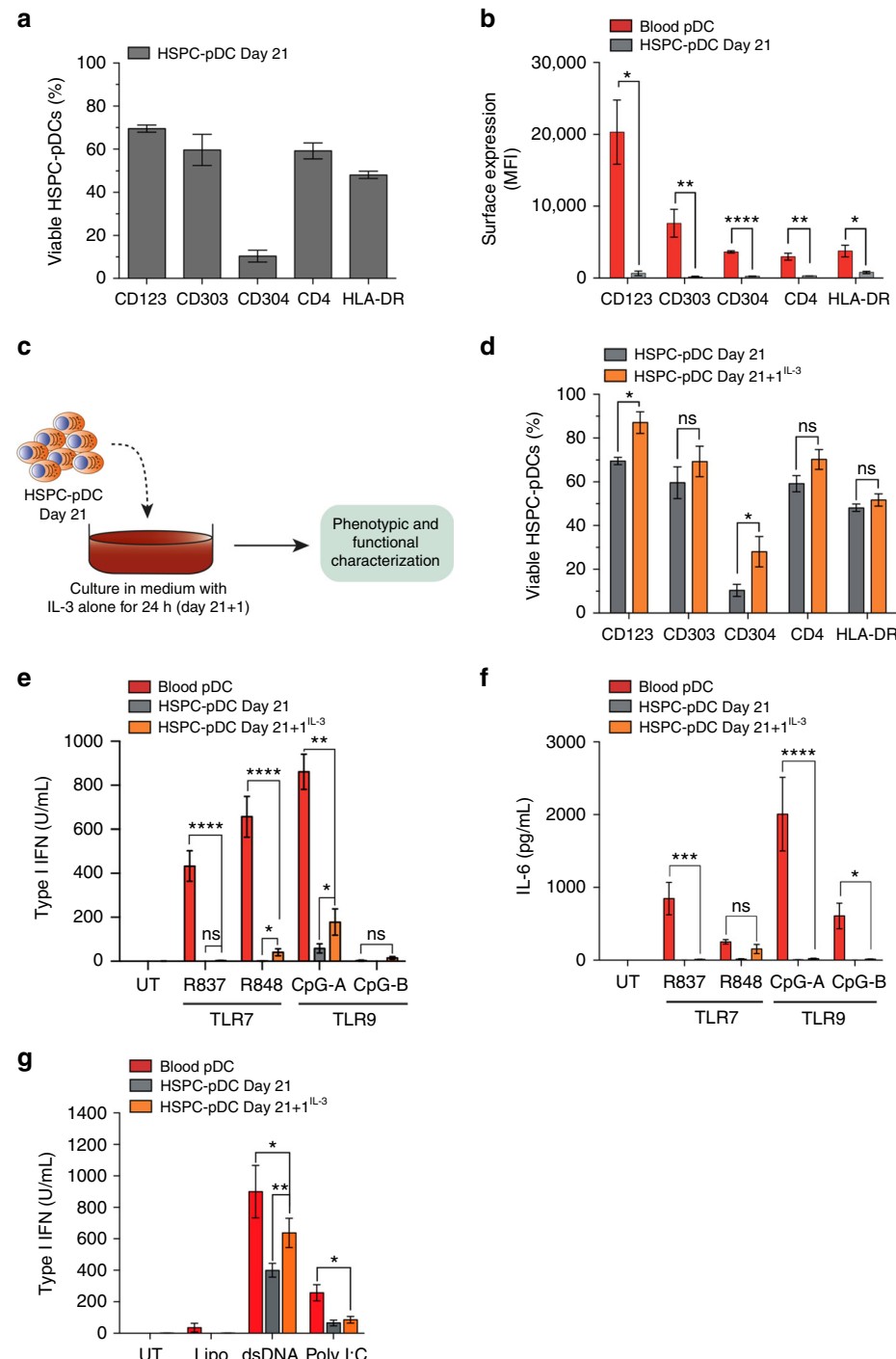

**Fig. 2** HSPC-pDCs elicit a precursor-pDCs phenotype. **a** Phenotypic analysis of pDC-associated markers on HSPC-pDCs at day 21 using Flow Cytometry. For gating strategy, see Supplementary Fig. 2a, b. **b** Surface expression levels of pDC-related markers on enriched HSPC-pDCs at day 21 in comparison to blood pDCs. **c** Graphical illustration of the experimental setup of removal of growth factors for 24 h after the 21-day differentiation. After enrichment of HSPC-pDCs at day 21, cells were depleted of growth factors (except IL-3). After 24 h (day 21 + 1) HSPC-pDCs were harvested for phenotypic or functional analyses. **d** Phenotypic comparison of HSPC-pDCs at day 21 or day 21 + 1. **e**, **f** Levels of type I IFN (**e**) or IL-6 (**f**) after stimulation with TLR7 or TLR9 agonists of enriched HSPC-pDCs cells at day 21, HSPC-pDCs Day 21 + 1$^{IL-3}$, or blood pDCs. **g** Levels of type I IFN after stimulation with agonists directed at cytosolic DNA and RNA sensors of enriched HSPC-pDCs cells at day 21, HSPC-pDCs Day 21 + 1$^{IL-3}$, or blood pDCs. Data shown are ± SEM of three donors. Statistical analysis was performed using regular two-way ANOVA (**b**, **d–g**) followed by Bonferroni post hoc test

pDCs represent a phenotypic and functional precursor to canonical pDCs, defining them as immature HSPC-pDCs.

**Interferon priming leads to a canonical pDC phenotype.** Recent studies have emphasized the role of IFNs in murine pDC development, homeostasis, and function[8–10,13]. Thus, we hypothesized that exogenous IFN may be required to drive HSPC-pDCs into a phenotype that more closely resembles blood pDCs and that these cells would then be able to respond efficiently to TLR agonists. To test this, we evaluated the effects of a 24 and

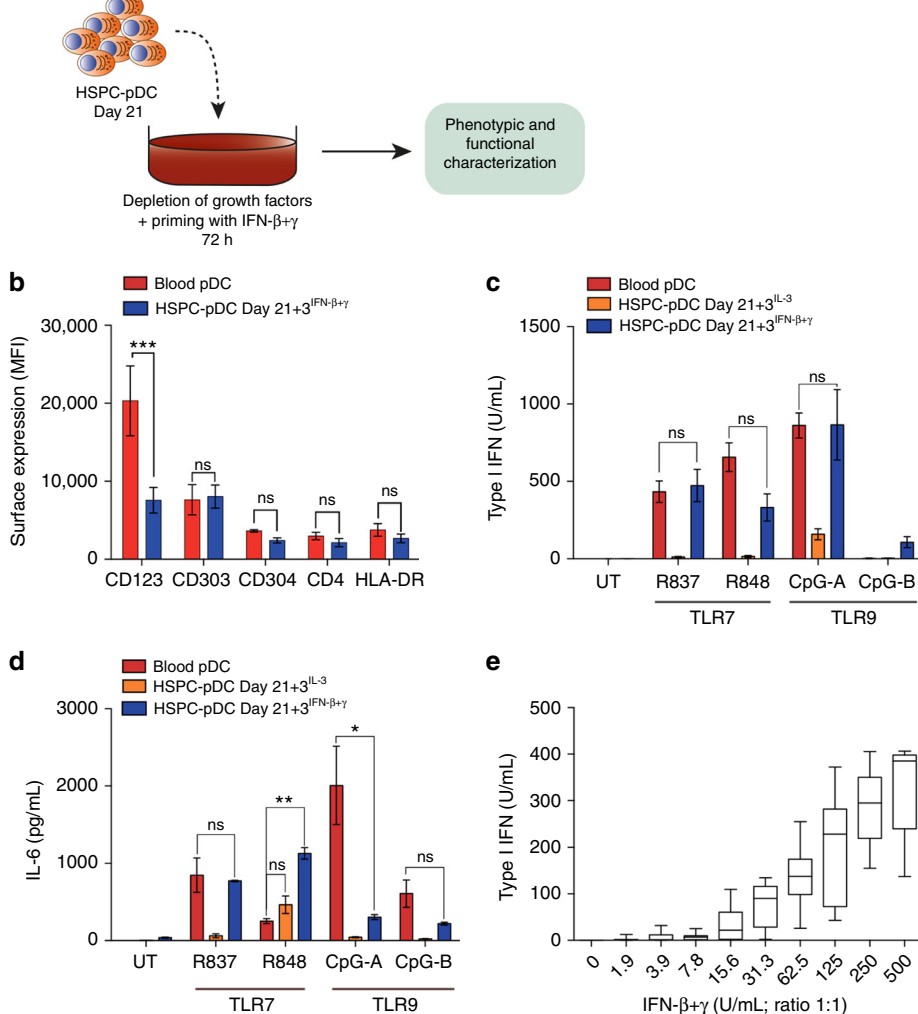

**Fig. 3** Upon IFN priming, HSPC-pDCs can be modulated to become classical pDCs. **a** Graphical illustration of the experiment setup of HSPC-pDC priming. After enrichment, HSPC-pDCs were primed in medium supplemented with IFN-β + γ or IL-3 alone. After 72 h (day 21 + 3) primed cells were harvested for phenotypic or functional analyses. **b** Surface expression levels of pDC-related markers on HSPC-pDC cells after priming for 72 h with IFN-β + γ (Day 21 + 3$^{IFN β+γ}$) and blood pDCs. **c**, **d** Type I IFN (**c**) or IL-6 (**d**) response of blood pDCs, unprimed or IFN-primed HSPC-pDCs cells to TLR7 and TLR9 agonists. **e** Type I IFN response of HSPC-pDCs primed with titrated concentrations of IFN-β + γ before being stimulated with the TLR7 agonist R837. Data shown are ± SEM of three donors. Statistical analysis was performed using regular two-way ANOVA (**b–d**) followed by Bonferroni post hoc test

72 h β- and/or γ-interferon exposure on HSPC-pDCs initiated at Day 21 (Day 21 + 1$^{IFN-β+γ}$ and Day 21 + 3$^{IFN-β+γ}$). Consistent with our hypothesis, we found that IFN-primed HSPC-pDCs acquired a surface phenotype profile that strongly resembles blood pDCs with no significant differences in the expression of CD303, CD304, CD4, or HLA-DR (Fig. 3a, b and Supplementary Fig. 4a–d). However, CD123 (IL-3 receptor) expression was still significantly lower, which has been reported to be caused by the supplementation of IL-3 during culture of pDCs[31]. Next, we challenged IFN-primed HSPC-pDCs with synthetic TLR7 and TLR9 agonists and evaluated the type I IFN and IL-6 responses. Intriguingly, we now observed that type I IFN and IL-6 reached levels comparable to that produced by stimulated blood pDCs (Fig. 3c, d and Supplementary Fig 5a–d). While the TLR7- and TLR9-mediated type I IFN responses were potent in IFN-primed HSPC-pDCs, the type I IFN response upon stimulation with synthetic agonists directed at cytosolic RNA and DNA sensors was less affected by IFN priming, indicating that IFNs mainly play a role in TLR7- and TLR9-dependent innate immune responses (Supplementary Fig. 6a, b). Notably, IFN-γ priming

predominantly promoted the secretion of IL-6, but was less effective at promoting a type I IFN response in comparison to IFN-β priming after synthetic TLR7 and TLR9 agonist stimulation (Supplementary Fig. 5a–d). Expression levels of CD303, CD304, and HLA-DR were increased by IFN-γ priming. Together, these data indicate that the signaling pathways of type I and II IFNs do not merely overlap, but rather lead to synergistic interferon-dependent signaling pathways (Supplementary Fig. 4a–d). By titrating in different amounts of exogenous IFN-β and IFN-γ, we found that HSPC-pDCs were able to acquire the phenotypic and functional traits of blood pDCs, using as little as 30 U/mL each of recombinant cytokine for priming (Fig. 3e and Supplementary Fig. 7a–e). To reduce potential negative effects of lot-to-lot variations in fetal calf serum, we also evaluated the phenotypic and functional appearance of HSPC-pDCs using commercially available serum-free media (SFEM II), which showed similar yield, pDC surface expression, and responses following TLR stimulation (Supplementary Fig. 8a–c).

Previous studies have identified rare pre-DC subsets, which, due to functional and phenotypic similarities to pDCs, may have

inadvertently been considered as mature pDCs in prior studies[32,33]. We therefore next evaluated if pre-DCs were present within the HSPC-pDC population. Pre-DCs are characterized as Lin[neg]CD123[+]CD303[+]CD304[+], but in addition express markers unrelated to pDCs, including CD2, CD5, and CD33. Furthermore, they secrete cytokines produced by cDCs such as IL-12 (refs.[32,33]). After immunomagnetic pDC enrichment on culture day 21, we observed that HSPC-pDCs were negative for the pre-DC marker CD33. Importantly, IFN priming of HSPC-pDCs did not lead to increased CD33 expression, indicating that IFN priming did not lead to generation of pre-DCs (Supplementary Fig. 9a, b). Notably, while it was possible to detect pre-DCs in a PBMC population, the subset was not detectable in the enriched blood pDC population, indicating that the enrichment procedure removes the blood pre-DC subset (Supplementary Fig. 9c, d). We further evaluated cell-surface expression of CD2, CD5, and CX3CR1, which have also been reported to define the pre-DC population. CD2 and CD5 were not expressed by HSPC-pDCs, neither at day 21 nor after IFN priming (Supplementary Fig. 9e, f). An increase in CX3CR1 expression was observed upon IFN priming, which reached an expression level similar to that of blood pDCs (Supplementary Fig. 9e, f). Finally, HSPC-pDCs at day 21 or IFN-primed HSPC-pDCs did not produce IL-12 upon stimulation with cDC-activating stimuli (R848, poly I:C or LPS), further supporting our conclusion that the previously described pre-DCs are not present within the HSPC-pDC population (Supplementary Fig. 10a–c).

Based on our observations that HSPC-pDCs from day 21 of culture displayed a previously undescribed precursor phenotype, we next searched for this population in vivo. We purified Lin[neg]CD11c[neg] cells from CB and analyzed the cells by flow cytometry for the expression of CD123 and CD303. Interestingly, we observed that fresh CB-pDCs contained two distinct populations, one with a mature phenotype (CD123[high]CD303[high]) and one with a precursor phenotype (CD123[dim]CD303[dim]) (Supplementary Fig. 11). Importantly, as observed for HSPC-pDCs, IFN priming increased the population of CD123[high]CD303[high] cells, indicating the presence of a pDC precursor phenotype in CB. However, a maturation was also evident during culture with IL-3 alone, albeit not to the same extent. A recent study argues that human pDCs are capable of self-priming by continuously secreting small amounts of type I IFN, enabling pDCs to retain a response through TLR7 and TLR9. This may partly explain the minor phenotypic shift we observed for unprimed cells[34]. Finally, we tested whether priming of HSPC-pDCs led to an altered response to TLR7 and TLR9 agonists via increased mRNA expression of genes involved in the TLR signaling pathway. The expression levels of IRF7 and TLR7 were significantly increased after IFN priming, whereas the expression of TLR9 remained unchanged (Supplementary Fig. 12). To test if exogenous priming of HSPC-pDCs initiated self-priming, we removed IFN-β and IFN-γ from HSPC-pDCs after one day in culture, and let the cells rest for an additional day in medium with IL-3 alone (Supplementary Fig. 13a). Priming of HSPC-pDCs led to increased expression of the IFN-α mRNA isotypes tested (α4, α16), and this expression was maintained after the resting phase, suggesting that self-priming occurs in this culture system (Supplementary Fig. 13b).

**HSPC-pDCs transcriptional responses**. To more comprehensively assess the transcriptional response of HSPC-pDCs to priming and TLR7 activation, we next performed whole transcriptome analysis. Cells from blood pDCs ($n = 4$) and primed HSPC-pDCs ($n = 4$) were mock-treated or subjected to 24 h of activation through TLR7. RNA was then immediately processed and analyzed by RNA-seq. Pathway analysis on genes differentially expressed between stimulated and non-stimulated cells was used to determine which biological processes were enriched during stimulation of blood pDCs and primed HSPC-pDCs (Fig. 4a). The most highly activated biological processes were shared between the two cell types, although the HSPC-pDCs tended to be more enriched for the shared pathways. When we examined the upstream regulators predicted to be responsible for inducing the transcriptional changes, we found a striking concordance between HSPC-pDCs and blood pDCs (Fig. 4b). Gene expression changes related to TLR7 signaling interactome (visualized in the outer circle) were strongly activated in both blood pDCs and HSPC-pDCs, supporting the fact that both cell types are fully capable of responding to the TLR7 agonist (Fig. 4c). The increased numbers of genes in the interactome of the HSPC-pDCs versus blood pDCs do indicate that a larger genetic shift from precursor HSCP-pDCs to fully TLR7-responsive pDCs is needed, supporting the fact that our day 21 HSCP-pDCs are less activated than unstimulated blood pDCs.

**HSPC-pDCs capacity to adopt an antigen-presenting phenotype**. An important but often neglected function of pDCs is their capacity to function as APCs. To test whether IFN priming affects the capacity of HSPC-pDCs to adopt a phenotype reminiscent of APCs, we next evaluated the expression of classical APC activation markers (CD40, CD80, CD83, and CD86)[1]. Unprimed as well as IFN-primed HSPC-pDCs treated with TLR7 agonist displayed comparable elevated expression of CD80 and CD83 (Fig. 5a and Supplementary Fig. 14). In contrast, expression of CD40 and CD86 was significantly higher in IFN-primed cells as compared to their unprimed counterparts.

To extend these observations, we next investigated if HSPC-pDCs were capable of activating T cells after pulsing them with CMV antigens. For this, we pulsed IFN-primed and TLR7 activated HSPC-pDCs with a cocktail of CMV-derived peptides (consisting of 14 peptides, each corresponding to a defined HLA class I-restricted T cell epitopes) and then co-cultured the cells with PBMCs derived from CMV-seropositive donors (Fig. 5b). First, to confirm that all three seroconverted donors responded to the CMV peptides, we merely used the APCs within the PBMC cultures to take up and present the CMV peptides to CD8[+] T cells (Fig. 5c). Next, we pulsed HSPC-pDCs with the CMV peptide cocktail and after extensive washing added them to the same three PBMC donors. Activation of CD8[+] T cells as measured by IFN-γ production was only evident when the co-cultured HSPC-pDCs had been pre-loaded with CMV (Fig. 5d). Together, these data indicate that activated HSPC-pDCs can present peptides and activate antigen-specific T cells.

A major drawback when using blood-derived pDCs is their short lifespan in culture, limiting the window of opportunity for in vitro evaluations. To estimate the lifespan of HSPC-pDCs in prolonged culture studies, we next compared viability of HSPC-pDC to blood pDCs over a prolonged culture of 12 days post differentiation, in medium with IL-3+/− IFN-β+γ. Within one day of isolation, only 50% of blood pDCs were viable. In contrast, HSPC-pDCs demonstrated superior survival during this period (Fig. 5e and Supplementary Fig. 15a–f). After 5 days of culture, the viability of HSPC-pDCs decreased to 60%. Notably, after 8 days of culture, only 10% of blood pDCs were viable, whereas unprimed HSPC-pDCs remained at 60% viability (Fig. 5e, f and Supplementary Fig. 15a–f). While viabilities were similar during the first 5 days between IFN-primed and unprimed HSPC-pDCs, a significant drop in viability was evident for IFN-primed cells from day 5 to 12 (Supplementary Fig. 15c).

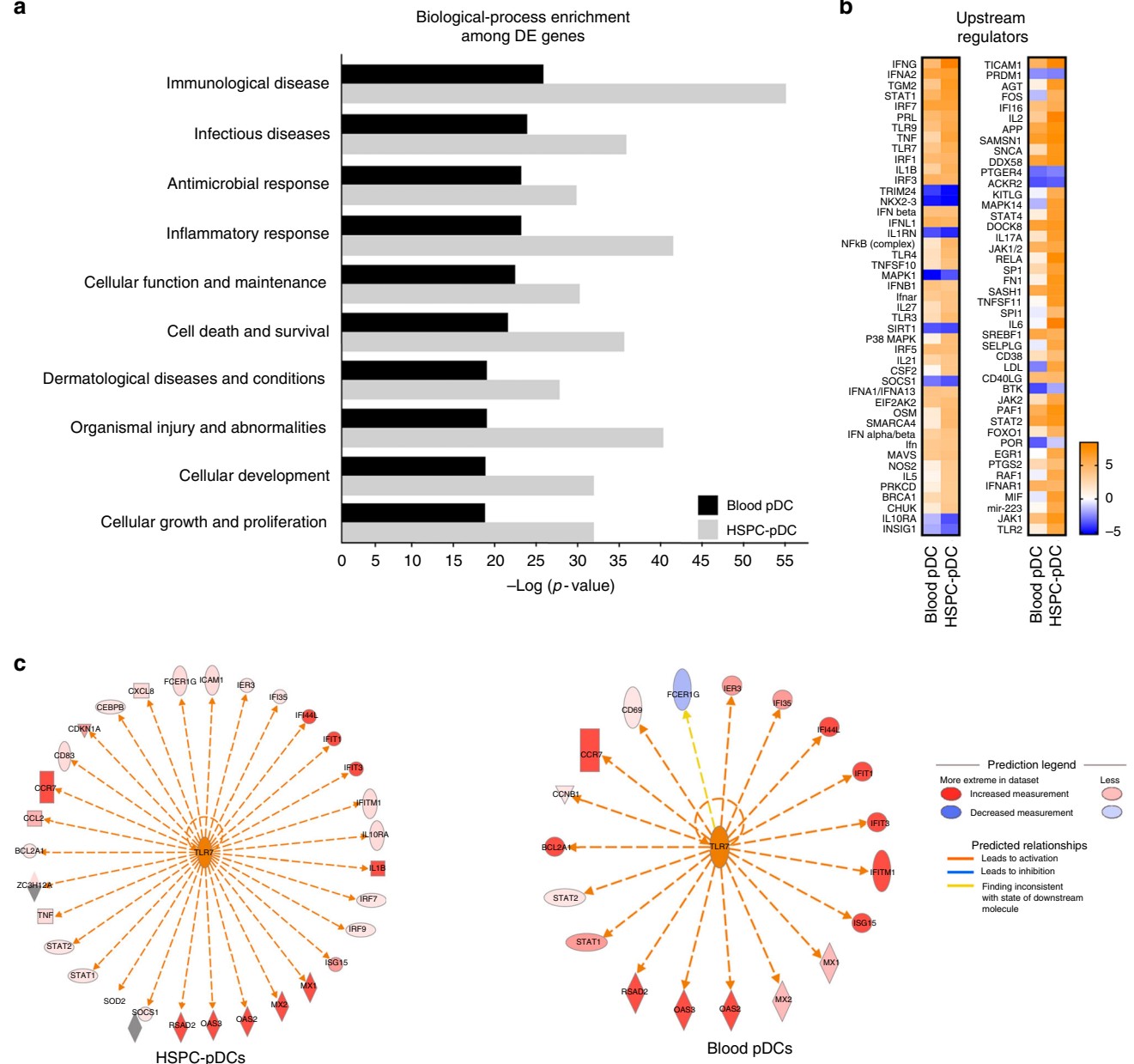

**Fig. 4** HSPC-pDCs and blood pDCs exhibit shared transcriptomic pathways. **a** Top 10 biological processes enriched in the set of differentially expressed genes and their associated *P* values for blood pDCs (black) and IFN-primed HSPC-pDC (gray) after 24 h stimulation with a TLR7 agonist. **b** Heat map (*z*-score) of both activated and inhibited upstream regulator genes within the top biological process corresponding to panel (**a**). **c** Illustration of regulated genes within the TLR7 pathway of after stimulation of blood pDCs and HSCP-pDC, respectively

**Production of genetically modified HSPC-pDCs.** To provide evidence that IFN signaling is essential for differentiation and/or maturation of HSPC-pDCs, we set out to genetically remove the IFN receptor. Whereas genetic manipulation of pDCs is currently not possible (discussed in the Introduction), recent advances in the CRISPR/Cas9 technology have resulted in successful and efficient gene editing of CD34[+] HSPCs[35–39]. Thus, we hypothesized that gene-edited pDCs could be generated successfully by initial modification of CD34[+] HSPCs and subsequent differentiation into pDCs. We employed a gene editing strategy where synthetic chemically modified sgRNAs and recombinant Cas9 protein, which forms a ribonucleoprotein (RNP) complex, were delivered to CD34[+] HSPCs by electroporation[35,36] (Fig. 6a). The sgRNAs were designed to target the open reading frame within

the first exon of *IFNAR1* (subunit of the type I IFN receptor) and MyD88. The latter target was included for its established role in TLR-mediated responses[1,10,34]. A previously published sgRNA targeting the safe harbor gene *CCR5* was used as a negative control[38]. A range of sgRNAs for *MyD88* and *IFNAR1* were initially screened for their potency to induce targeted gene disruption by insertions and deletions (Indels) by plasmid electroporation in K562 cells (Supplementary Fig. 16a). The three sgRNAs with the highest efficacy for each target were then tested as synthetic sgRNAs with chemically modified nucleotides at both termini by Cas9 RNP delivery by electroporation of CD34[+] HSPCs, leading to the selection of a single potent sgRNA for each target gene yielding 73.4 ± 4.5% and 86.7 ± 2.7% Indel frequencies for *MyD88* and *IFNAR1*, respectively (Supplementary Fig. 16b).

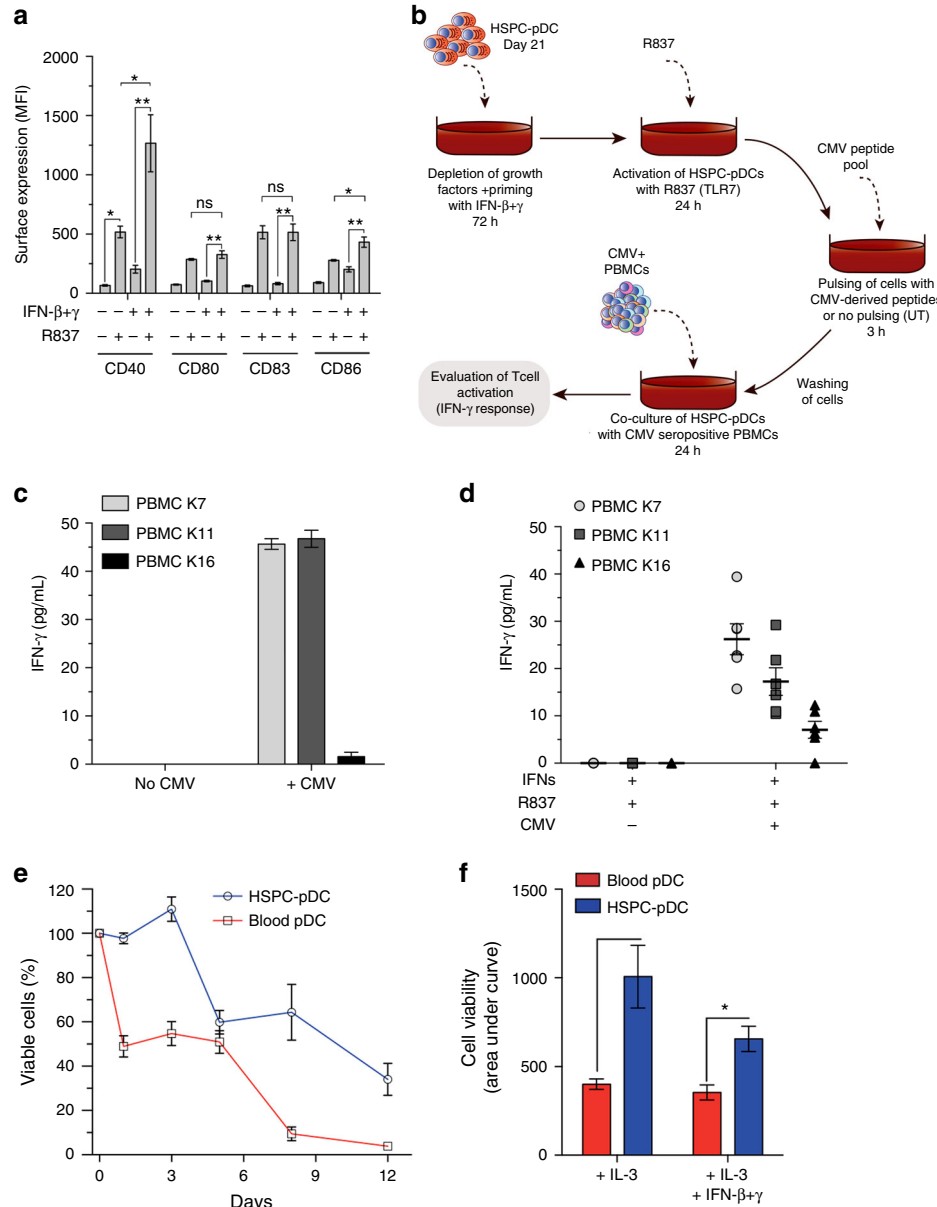

**Fig. 5** HSPC-pDCs have the capacity to adopt an antigen-presenting phenotype. **a** Expression levels of the indicated maturation APC markers on IFN-primed/unprimed HSPC-pDCs at day 21 + 3 after stimulation with the TLR7 agonist R837. **b** Graphical illustration of the experimental setup for T cell stimulatory capacity of HSPC-pDCs. HSPC-pDCs (IFN-primed and TLR7-activated) were pulsed with a mix of peptides derived from CMV (CMV ProMix) or left untreated (UT). After 3 h, cells were washed and co-cultured with allogeneic PBMCs harvested from CMV-seropositive donors and T cell activity was subsequently evaluated by measuring levels of IFN-γ in the supernatant 24 h later. **c** Levels of IFN-γ produced by PBMCs from three CMV-seropositive donors after co-culture with pDCs loaded with CMV ProMix. **d** Levels of IFN-γ produced by the same three PBMC donors mixed in a 10:1 ratio with three HSPC-pDC donors (IFN-primed and TLR7-activated) for 3 h with or without a pulse of CMV ProMix. **e** Blood pDCs or HSPC-pDCs were cultivated for 1, 3, 5, 8, and 12 days in medium supplemented with IL-3. At each time point, the viability was assessed by flow cytometry. **f** Comparison of area under curve of HSPC-pDCs and blood pDCs in the different conditions tested. Data shown are ±SEM of three (**a–d**) or four (**e–f**) donors. Statistical analysis was performed using regular two-way ANOVA (**e**, **f**) followed by Bonferroni post hoc test

The two selected sgRNAs and the *CCR5* control sgRNA were then used in CD34[+] HSPCs that following a 3-day recovery phase after electroporation were differentiated into pDCs (Fig. 6a, b). Importantly, no change in the proliferation rate and total cell yield was observed during differentiation (comparing Figs. 1b and 6b) and high Indel rates were observed pre- and post differentiation (Fig. 6c and Supplementary Fig. 16c, d). When we evaluated the overall yield of HSPC-pDCs on day 21 we observed a ~4-fold decrease in yield compared to previous experiments. We ascribed this to either the nucleoporation procedure itself, as mock-

electroporated cells were equally affected, or to the increased period of cultivation (Fig. 6d). Analysis of Indel frequencies after immunomagnetic pDC enrichment confirmed that HSPC-pDCs had similar levels of Indels as the total non-enriched cell population (Supplementary Fig 16d), and MyD88 protein levels determined by western blotting confirmed potent knock-down (Fig. 6e). IFNAR1-targeted HSPC-pDCs showed reduced capacity to induce downstream signaling through IFNAR, as stimulation with IFN-β failed to induce STAT1 phosphorylation, whereas signaling through the IFN-γ receptor was unaffected (Fig. 6e and

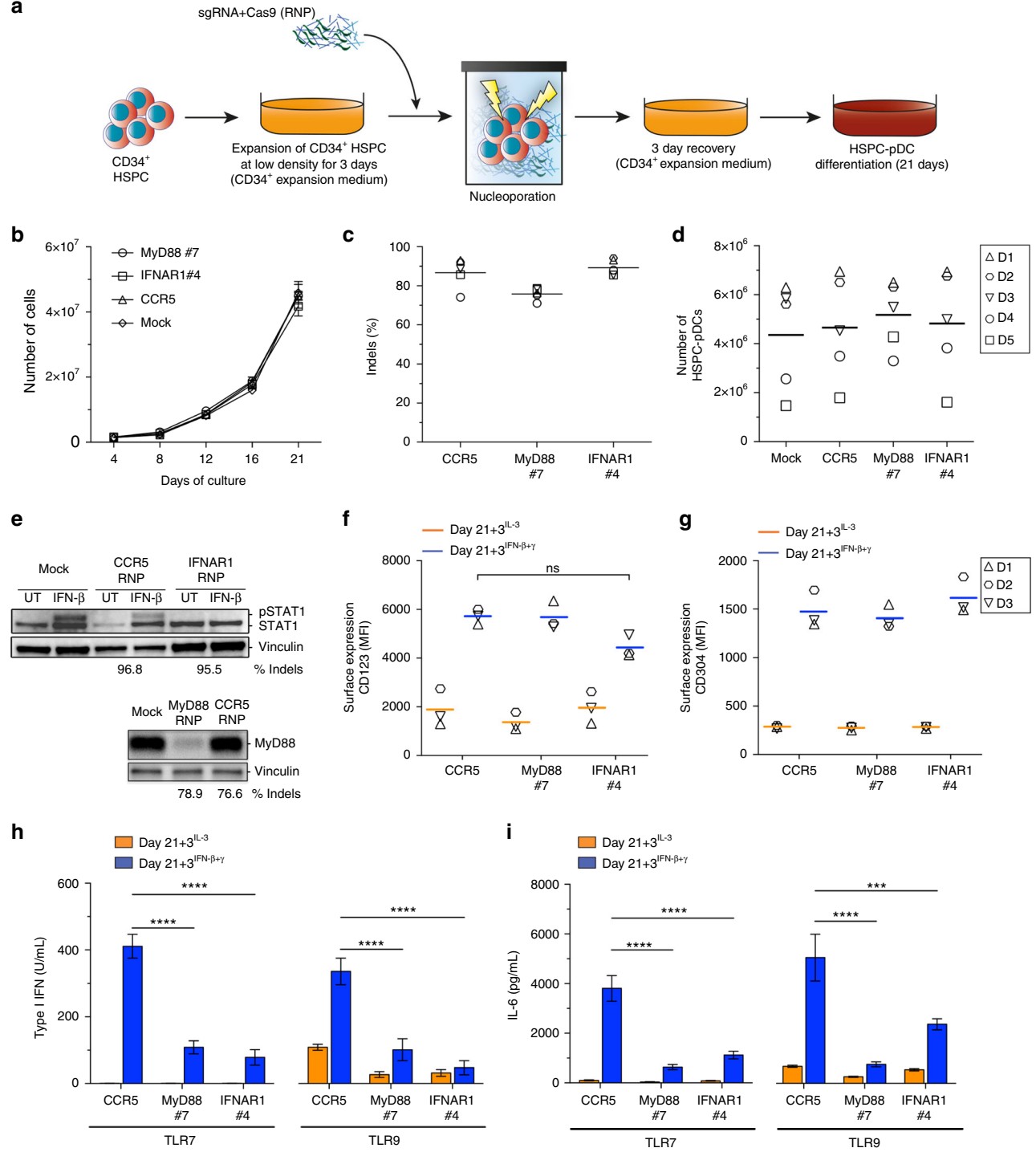

**Fig. 6** HSPC-pDCs are amenable to genetic modifications. **a** Graphical illustration of the protocol for CRISPR/Cas9-mediated gene editing using Cas9 RNP. CD34[+] HSPCs were initially cultured in CD34[+] HSPC medium at low density for 3 days before being electroporated with Cas9 RNP complexes. After a 3-day recovery phase, pDC differentiation was initiated. **b** Expansion of cells during differentiation from HSPCs into HSPC-pDCs. **c** Indel frequencies after the 3-day recovery phase quantified by TIDE analysis for cells targeted at *MyD88*, *IFNAR1*, and *CCR5* (control). **d** Numbers of HSPC-pDCs at day 21 for gene-edited HSPC-pDCs. **e** Western blot analysis showing induction of pSTAT1 upon 3 h IFN-β stimulation (upper panel) and protein levels of MyD88 (lower panel) in HSPC-pDCs targeted at *IFNAR1* and *MyD88*, respectively (detection Ab for STAT1: CST 9172). Indel frequencies at the respective targets are listed below the western blot. **f**, **g** Surface expression levels (MFI) of CD123 (**f**) and CD304 (**g**) in IFN-primed and unprimed HSPC-pDCs with gene editing at *MyD88*, *IFNAR1*, and *CCR5*. **h** Functional levels of type I IFN after stimulation with R837 (TLR7) or CpG-A (TLR9) in unprimed and IFN-primed HSPC-pDCs gene-edited at *CCR5*, *MyD88*, or *CCR5*. **i** Levels of IL-6 after stimulation with R837 (TLR7) or CpG-A (TLR9) in unprimed and IFN-primed HSPC-pDCs targeted at *CCR5*, *MyD88*, or *IFNAR1*. Data are ±SEM of five (**b**–**d**) or three (**f**–**i**) donors. Statistical analysis was performed using regular two-way ANOVA followed by Bonferroni post hoc test. Two additional donors are shown in Supplementary Fig. 17

Supplementary Fig. 16e, f). Next, the HSPC-pDCs were analyzed for two pDC phenotypic markers (CD123 and CD304) and for functionality after IFN priming. CRISPR/Cas9-treated cells showed expression of both surface markers and expression of both markers increased markedly following IFN priming, which recapitulates earlier observations (Fig. 6f, g and Supplementary Fig. 17a, b).

*MyD88*-targeted HSPC-pDCs were then IFN-primed and stimulated with TLR7/9 agonists. In both cases of TLR stimulation, type I IFN and IL-6 induction were significantly decreased in the KO cells (Fig. 6h, i and Supplementary Fig.17c). Interestingly, when we evaluated the response to TLR agonists in HSPC-pDCs targeted at the *IFNAR1* gene, we found that the cells displayed a decreased capacity to induce type I IFN, supporting our previous findings that type I IFN feedback is a prerequisite for TLR7- and TLR9-dependent type I IFN induction (Fig. 6h). Moreover, IL-6 production after stimulation was also diminished in IFNAR1 sgRNA-targeted HSPC-pDCs, indicating type I IFN signaling is important for TLR7 and TLR9 activity in pDCs (Fig. 6i). Of note, in some donors we observed a full abrogation of the type I IFN response, despite ~20% of the cells still possessing the WT allele of IFNAR1 (as estimated from the TIDE analysis), indicating an amplification loop may be required for the TLR7/9-mediated type I IFN response to occur in the remaining cells (Fig. 6h, i and Supplementary Fig. 17d). In summary, our results identify a hitherto unknown role of interferons in promoting the activation of human precursor pDCs into a phenotype that strongly resembles that of pDCs derived from peripheral blood. We also demonstrate that gene-edited HSPC-pDCs can be produced by initially modifying CD34[+] HSPC before differentiation to HSPC-pDCs. Using this system, we confirm that while MyD88 and IFNAR1 do not appear to be important in human pre-pDC development, they are essential for functional pDC immune responses upon stimulation through the TLR pathway.

## Discussion

Here we present the first evidence that IFN priming is essential for the ability of pDCs to mount a robust and efficient innate immune response. We show this using a novel in vitro approach to expand and differentiate human CD34[+] HSPCs, first into precursor pDCs, and then into fully mature and functional pDCs. The power of the system is found within its simplicity, the increased survival of the pDC, high cell yields, and phenotypic and functional resemblance of these HSPC-derived cells to blood-collected pDCs. In addition, the approach is readily amenable for genetic manipulation by CRISPR/Cas9 approaches. This is an important benefit of this culture system, since inhibition of gene expression in by RNAi-based methods in pDCs can be problematic due to non-specific immune activation through, e.g. RIG-I or TLR3/7/8[15,40–42]. Furthermore, pDCs in general have been difficult to genetically modify, and even siRNAs engineered to be less immuno-stimulatory, such as 2′O-methyl modifications, inadvertently antagonizes TLR7, which affects downstream assays[43–45]. Moreover, designing a non-immuno-stimulatory siRNA that retains RNAi potency is often difficult and requires trial and error[46]. Our demonstration that CRISPR/Cas9-based gene editing can be accomplished in human pDCs will lay the groundwork for future elucidation of specific molecular pathways involved in pDC development and function. We also note that our use of a transient RNP-based delivery system in early developing HSPCs will reduce the risk of potential off-target effects that are more prominent during continuous expression of a gene cassette (e.g. lentiviral transduction) also used in CRISPR/Cas9-based approaches.

While previous culture systems have allowed for the generation of pDCs, our work identified a limitation of those studies, namely that such previously generated HSPC-pDCs display a precursor phenotype, that must be overcome in order to generate pDCs that resemble the majority of pDCs found in the bloodstream. Type I IFN responses in in vitro-generated pDC after stimulation with the synthetic TLR9 agonist CpG-A have previously been reported, while synthetic TLR7 agonists, to our knowledge, have never been tested[20,21,23–25]. Moreover, limited comparisons to blood pDCs have been conducted[20,21,23–25]. We found that unprimed HSPC-pDCs produced type I IFN in response to CpG-A, although at significantly reduced levels compared to blood pDCs. In addition, no type I IFN responses were observed upon stimulation of HSPC-pDCs with synthetic TLR7 agonists. Notably, the observed precursor state of unprimed HSPC-pDCs shares many of the intrinsic features of neonatal pDCs, including a lack of a TLR7-mediated response and decreased surface expression of pDC markers[47–50].

Of note, three distinct maturation stages of pDCs have recently been identified by Martín-Martín et al.[51] within adult bone marrow. These stages were separated by their expression levels of pDC phenotypic markers (CD123, CD303, and CD304), and the capacity to produce type I IFN upon synthetic TLR9 agonist stimulation[51]. Martín et al. found that pDCs from peripheral blood produced markedly higher levels of type I IFN in comparison to bone marrow-derived pDCs. We identified two populations of lin[neg]CD11c[neg] cells within CB separated by their expression levels of the pDC markers CD123 and CD303. Importantly, IFN priming increased the fraction of cells with high expression levels of these pDC markers, whereas the population with intermediate expression decreased. Of note, an increase in the CD123[high]CD303[high] population was observed upon IL-3 conditioning alone, suggesting that self-priming capabilities exist which can potentially overcome the need of exogenous priming with IFN[34]. Overall, this indicates that pDCs undergo different stages of development before reaching phenotypic and functional maturity.

Our findings indicate that immature pDCs require the withdrawal of precursor-supporting growth factors and the addition of a positive factor in order to attain activity and respond to pathogens. Infection and other physiologic stresses are known to stimulate cytokine output, in particular type I and II IFNs, both regionally and systemically, which would support local priming of pDCs and their trafficking to the site of infection[5–7]. While IFNs have traditionally been considered to be antiviral, pro-inflammatory, and pro-apoptotic, they have recently also been shown to regulate survival of activated T cells and promote proliferation of dormant HSPCs via a STAT1-dependent pathway[7,52]. Similarly, Chen et al. reported that IFNAR1[−/−] and STAT2[−/−] mice display significantly decreased numbers of pDCs within the blood, supporting our hypothesis that IFN priming is involved in pDC homeostasis[8–10]. Moreover, as demonstrated by Vollstedt et al.[8] murine pDCs are mobilized in neonatal mice upon IFN-α treatment. Although a direct role of type II IFNs in pDC development and activation remains to be elucidated, both type I and II IFNs are known to signal through a JAK–STAT signaling pathway. Accordingly, it is well-established that many ISGs are regulated by both factors[53]. However, other genes are known to be selectively regulated by the two types of IFN, which is likely regulated by differential activation of different STAT proteins[53]. Notably, both type I and II IFNs induce STAT3 phosphorylation, which is known to induce the transcription factor E2-2, which is directly involved in pDC development by inducing a number of pDC-specific genes, including Spi-B, IRF7, TLR7, TLR9, ILT7 (CD85g), and CD303 (refs.[53–56]). Accordingly, we observed up-regulation

of pDC-related markers, including CD123, CD303, and CD304, upon priming with IFN-β and/or IFN-γ. Furthermore, from the transcriptomic analysis we saw that priming of HSPC-pDCs with IFNs resulted in a shared pattern of differentially expressed genes activating biological pathways as that observed upon stimulation of blood pDC, suggesting the existence of shared functional responses in the blood- vs. HSPC-derived pDCs.

Interestingly, we found IFN-γ to be most potent in upregulating pDC-associated surface markers, which may suggest this cytokine has a higher preference for STAT3-mediated signaling. However, we found that IFNAR knockdown in HSPC-pDCs abrogated type I IFN responses to TLR7 and 9 agonists, suggesting that type I IFN feedback is crucial for the TLR function of pDCs. As type I IFN indirectly signals the presence of a pathogen, this may in turn serve to regulate the function of the receptors, so aberrant immune activity is avoided. This is analogously supported by the concept that many proteins implicated in TLR7 and TLR9 processing or signaling pathways are IFN-regulated, including IRF7, PLSCR1, and viperin[57–60].

While the role of IFNs in human pDC homeostasis remains to be fully elucidated, it is known that pDCs are involved in the immunopathogenesis of several autoimmune diseases, including psoriasis, type I diabetes (T1D), and SLE. These diseases are all characterized by a distinct "type I IFN signature", and increased Th1-mediated type II IFN responses, which is believed to have a major role in the instigation and severity of disease[61–67]. Importantly, pDCs are significantly increased within peripheral blood of patients with SLE and T1D, and in psoriasis patients, pDCs are recruited to psoriatic plaques[64–66,68]. Moreover, previous viral infections or IFN-α therapy during hepatitis C infection are known to promote the development of T1D[61–65]. Taken together, these studies indicate that a type I and II IFN environment observed during autoimmune diseases or viral infections serve to mobilize pDCs from the bone marrow. With our findings that both type I and II IFNs are critical for HSPC-pDCs to achieve a functional and phenotypic resemblance to blood pDCs, our study adds further information on how such factors affect the functionality of pDCs.

In conclusion, our study demonstrates a previously undescribed crucial role of IFNs in the maturation of human pDCs. These studies apply a novel in vitro differentiation protocol, and resulted in the identification of a two-step developmental process of pDCs that entails the initial generation of immature pre-pDCs that display a poor innate immune response through the classical endosomal TLR7/9 receptors, followed by subsequent priming by type I and II IFNs that rapidly matures the cells into a fully functional phenotype. We believe that this finding will have important implications for the understanding of a range of immune-related disorders.

## Methods

**Ex vivo generation of pDCs from CD34+ HSPC.** CD34+ HSPCs were purified from human umbilical cord blood (CB) acquired from donors under informed consent via either the Binns Program for Cord Blood Research at Stanford University or from Department of Gynecology and Obstetrics, Aarhus University Hospital, Aarhus. Briefly, mononucleated cells were recovered by standard Ficoll-Hypaque (GE Healthcare) density-gradient centrifugation. CD34+ cells were subsequently isolated using anti-CD34 immunomagnetic beads (positive selection) following the manufacturer's instructions (EasySep™ Human cord blood CD34+ positive selection kit, STEMCELL Technologies or CD34 MicroBead Kit UltraPure, human, Miltenyi Biotec). CD34+ HSPC were either freshly used or cryo-preserved until future use. For HSPC to pDC differentiation, CD34+ HSPC were plated in 48-well plates in RPMI-1640 medium (Lonza), supplemented with 10% heat-inactivated fetal calf serum (FCS) (HyClone®), 600 μg/mL L-glutamine (Sigma), 200 U/mL penicillin, and 100 μg/mL streptomycin (Gibco®, Life Technologies). Notably, we also found it is possible to substitute the culturing protocol with serum-free conditions, using SFEM II (STEMCELL Technologies) supplemented with 20 μg/mL streptomycin and cytokines/growth factors. Furthermore, the cytokines Flt3-L, TPO, and IL-3 (Peprotech) were added as a baseline. In addition

to the baseline, the cytokines SCF, IL-7 (Peprotech), and the chemical StemRegenin 1 (STEMCELL Technologies) were tested individually or in combination at various concentrations. Cells were cultured at 37 °C, 95% humidity, and 5% CO₂ for 21 days. Medium was refreshed every 4 days with medium containing the designated cytokines. Total cell numbers during the cultivation period were determined using the cell counter Moxi Z™ (ORFLO Technologies) or TC20 (Bio-Rad). At day 21 HSPC-pDCs were enriched using a negative selection kit that depletes non-pDCs, according to the manufacturer's protocol (EasySep™ Human Plasmacytoid DC Enrichment kit, STEMCELL Technologies).

**pDC enrichment from peripheral blood.** PBMCs were isolated by Ficoll-Hypaque density centrifugation of normal healthy blood donor buffy coats obtained from Aarhus University Hospital Blood Bank. Blood pDCs were enriched by negative selection following the instructions by manufacturer (EasySep™ Human Plasmacytoid DC Enrichment kit; STEMCELL Technologies).

**Generation of monocyte-derived dendritic cells (mo-DCs).** CD14+ monocytes were initially purified from PBMCs by negative immunomagnetic selection according to the manufacturer's instructions (EasySep™ Human Monocyte Enrichment kit, STEMCELL Technologies). Purified monocytes were washed twice and cultured in six-well plates at a density of 0.5 × 10⁶ cells/mL in 1.2 mL of RPMI-1640 medium (Lonza), supplemented with 2% normal human serum type AB (NHS AB), 600 μg/mL L-glutamine (Sigma), and 20 μg/mL gentamicin (Gibco). Cultures were established in the presence of 100 ng/mL human GM-CSF (Peprotech) and 20 ng/mL IL-4 (Peprotech). Cells were cultured at 37 °C, 95% humidity, and 5% CO₂ for 6 days. During the cultivation, cultures were supplemented with fresh medium supplemented with GM-CSF and IL-4. At day 6, cells were functionally evaluated.

**Priming of HSPC-pDCs.** HSPC-pDCs were cultured in RPMI-1640 medium with 10% heat-inactivated FCS, 600 μg/mL L-glutamine, 200 U/mL penicillin, and 100 μg/mL streptomycin, but depleted for growth factors used for the differentiation process described above. As described in individual figure legends, medium was supplemented with 250 U/mL IFN-β (PBL Assay Science), 250 U/mL IFN-γ (PeproTech), or both cytokines simultaneously (IFN-β + γ). After 24 (day 21 + 1) or 72 (day 21 + 3) hours, cells were washed in warm medium, and phenotypically or functionally characterized.

**Cell stimulation.** For functional characterization, cells were seeded in 96-well plates. Cells were left untreated or stimulated with stimuli as shown in Supplementary Table 1. During stimulation cells were kept in medium with 20 ng/mL IL-3. Twenty hours post stimulation supernatants were harvested and kept at −20 °C until cytokine quantifications. For phenotypic characterization, IFN-primed or unprimed cells were seeded in 48-well plates and stimulated for 24 h before phenotypic analysis by flow cytometry.

**Measurement of functional type I IFN.** Bioactive functional type I IFN was quantified in supernatants using the reporter cell line HEK-Blue™ IFN-α/β (Invivogen). The cell line was maintained in DMEM + GlutaMax™-I (Gibco®, Life Technologies), supplemented with 10% heat-inactivated FCS, 100 μg/mL streptomycin and 200 U/mL penicillin, 100 μg/mL normocin (InvivoGen), 30 μg/mL blasticidin (InvivoGen), and 100 μg/mL zeocin (InvivoGen). Cells were passaged using 1× trypsin (Gibco®, Life Technologies). To ensure optimal type I IFN response and stable expression of the plasmids encoding hIRF9, hSTAT2, and SEAP, cells were not passaged more than 20 times. For measurement of functional type IFN, cells were seeded at 3 × 10⁴ cells/well in 96-well plates in 150 μL medium. Cells were grown as previously described, but without blasticidin and zeocin. The following day, 50 μL of supernatant from stimulated cells or a type I IFN standard range was added to the cells. After 24 h of incubation, 20 μL of supernatant from the HEK-Blue cells was subsequently added to 180 μL QUANTI-Blue™. SEAP activity was assessed by measuring optical density (OD) at 620 nm on a microplate reader (ELx808, BioTEK). The standard range was made with IFN-α (IFNa2 PBL Assay Science) and ranged from 2 to 500 U/mL.

**Enzyme-linked immunosorbent assay.** Protein levels of IL-6, IL-12 p70, or IFN-γ in supernatants were evaluated using ELISA kits for IL-6 (Biolegend), IL-12 (Human IL-12 p70 DuoSet ELISA kit, R&D Systems), and IFN-γ (Human IFN-γ DuoSet ELISA kit, R&D Systems), following the manufacturer's instructions.

**Western blot.** To evaluate protein levels, cells were lysed in Pierce RIPA buffer (Thermo Scientific) supplemented with 10 mM NaF, 0.2% SDS, 1× protease inhibitor (Roche), and 1× XT sample buffer (Bio-Rad). Whole-cell lysates were heated at 95 °C for 5 min before being separated on a 10% SDS Bis-Tris Gel (Criterion TGX gels, Bio-Rad) and blotted onto PVDF membrane (Bio-Rad). Proteins were stained using primary antibodies (1:1000) directed against MyD88 (D80F5; Cell Signaling Technology), STAT1 (Cell Signaling Technology 9172), pSTAT1 (Tyr701, D4A7; Cell Signaling Technology, 7649), and vinculin (hVIN-1; Sigma-Aldrich) and secondary peroxidase-conjugated Affinipure F(abr)2 donkey

anti-rabbit (IgG) (1:15000) (Jackson Immuno Research). Proteins were detected using Clarity Western ECL Blotting Substrate (Bio-Rad) followed by visualization with an ImageQuant LAS4000 mini biomolecular imager (GE Healthcare). Uncropped scans of important blots are included in the Supplementary Figure 18.

**Morphology of cells**. To examine the morphology of cells at day 21, cells were smeared out on glass slides and fixed with methanol. May–Grünwald Giemsa Staining (Merck) was performed according to the manufacturer's protocol. Slides were examined using a Leica DM IL LED microscope.

**Phenotypic analysis with flow cytometry**. Immunophenotypical analysis was performed using flow cytometry. Briefly, $2 \times 10^5$ cells were spun down and resuspended in staining buffer (PBS with 0.5% BSA and 0.09% NaN$_3$). Cells were blocked on ice for 30 min before being stained with fluorochrome-conjugated antibodies for 30 min. After three washing steps in staining buffer, cells were fixed in 0.9% formaldehyde. Fluorescence intensities were measured by flow cytometry either on an LSR Fortessa flow cytometer with four lasers (405, 488, 561, and 640 nm) and 18 photomultiplier (PMT) detectors (Becton Dickinson), a NovoCyte Analyzer equipped with three lasers (405, 488, and 640 nm) and 13 PMT detectors (ACEA Biosciences, Inc), or on a CytoFlex (Beckman Coulter) with four lasers (405, 488, 561, and 633 nm) and 14 PMT detectors. Antibodies used to determine phenotype of cells, as well as working concentrations, are listed in Supplementary table 2. Data were analyzed using FlowJo software (version 10, Tree Star, Ashland, OR, USA). Individual gating strategies are depicted in relevant figures and outlined in figure legends. FMO controls were utilized to distinguish different populations.

**Viability over prolonged culture**. To assess the viability over prolonged culture of HSPC-pDCs or blood pDCs, cells were collected at each time point and subsequently analyzed using flow cytometry. Cells were stained with Zombie aqua as a viability dye, and viable cell counts were measured and normalized to a fixed number of counting beads (CytoCount beads; Dako).

**Quantitative real-time PCR**. Gene expression levels of IFN-α2, IFN-α4, IFN-α16, IRF7, TLR7, and TLR9 were determined by reverse transcription (RT)-qPCR using TaqMan detection systems (Life Technologies). Expression levels were normalized to DAG1, and data are presented as relative expression levels. The following TaqMan assays were used: IFN-α2; Hs00265051_s1, IFN-α4; Hs01681284_sH, IFN-α16; Hs03005057_sH, TLR7; Hs01933259_s1, TLR9; Hs00370913_s1, DAG1; Hs00189308_m1.

**Antigen presentation assays**. Enriched HSPC-pDCs were primed with IFN-β + γ for 72 h and subsequently stimulated with R837 (Invivogene, 2.5 μg/mL). After 24 h stimulation, HSPC-pDCs were pulsed with 1 μg per peptide/mL of CMV (Pro-Mix$^{TM}$ CMV Peptide Pool, ProImmune) or left untreated. After 3 h cells were washed and subsequently co-cultured with PBMCs from CMV-seropositive donors. As controls, PBMCs from the same donors were stimulated with 1 μg per peptide/mL of CMV or left untreated. PBMCs and HSPC-pDCs were co-cultured at a cell ratio of 10:1 ($10^5$ PBMC and $10^4$ HSPC-pDCs) in duplicates in 96-well plates, 200 μL 10% RPMI supplemented with 20 ng/mL IL-3. After 24 h incubation supernatants T cell activation was evaluated by measuring IFN-γ levels in supernatants.

**RNA-seq analysis**. Library preparation was performed using 1 μg of total RNA from each sample, using polyA selection. After fragmentation, first strand cDNA synthesis was performed using random hexamers followed by second strand cDNA synthesis. After double-stranded cDNA purification using AMPure XP beads, the fragments were end-repaired, 3′ adenylated, and NEBNext Adaptor with hairpin loop structure was ligated. cDNA fragments of preferentially 350 bp in length were selected using the AMPure XP system and PCR amplification was performed to create the final cDNA library. The library was sequenced using Illumina 150 bp paired-end reads (~30M reads/sample). Library preparation and sequencing was conducted by Novogene (Sacramento, USA). Transcriptome analysis was performed using Partek Flow built 5.0.16.1128 (St. Louis, USA). Adaptor-trimmed reads were imported into Partek Flow and aligned to the human reference genome hg38 using STAR (version 2.5.3a). The aligned reads were annotated to transcriptome (hg38 RefSeq Transcripts 84) using the Partek E/M annotation model with strict paired-end compatibility required. The aligned reads were normalized using TPM (transcripts per kilobase million). Genes with extremely low expression were filtered out based on a minimal expression cutoff of 1.0 TPM across samples; 4732 genes were included in downstream analyses. Differential gene expression was calculated using ANOVA in Partek Flow with treatment as a fixed factor and participant ID as a random factor, as some cell populations were from the same donor. Lists of differentially expressed genes were generated by using the following thresholds: absolute fold change >5 and FDR <0.05. Pathway analysis and identification of predicted upstream regulators was performed using Ingenuity Pathway Analysis (IPA) (Qiagen, Redwood City, USA). The calculated $z$-score predicts the direction of change (positive $z$-score indicates increased activation state of

predicted upstream regulators (orange), and negative value indicates a decrease in activation state (blue).

**Screening of sgRNAs**. sgRNAs were initially screened for the capacity to induce insertions or deletions (Indels) in the immortalized K562 cell line (ATCC). K565 (ATCC) was maintained in RPMI-1640 (Lonza) supplemented with 10% FCS (HyClone®), 100 μg/mL streptomycin and 200 U/mL penicillin, and 2 mM ʟ-glutamine. Two annealed oligonucleotides carrying the sgRNA target sequences were cloned into pX330 (pX330-U6-Chimeric_BB-CBh-hSpCas9 was a gift from Feng Zhang, Addgene plasmid #42230) as previously described[69]. The pX330 plasmid contains a human U6 promotor driving the expression of the sgRNA and a SpCas9 expression cassette (for sgRNA sequences see Supplementary Table 3). Delivery of sgRNA-px330 plasmid to cells was performed by nucleoporation using the Lonza 4D-Nucleofector$^{TM}$ System (program FF-123). Three days after transfection, Indel frequencies were quantified using TIDE (Tracking of Indels by Decomposition)[70]; Genomic DNA was extracted and PCR amplicons spanning the sgRNA target site were generated (for primers see Supplementary Table 4). Purified PCR products were then Sanger-sequenced and Indel frequencies quantified using the TIDE software (http://tide.nki.nl). A reference sequence (mock-transfected sample) was used as a control.

**Making genetically modified HSPC-pDCs**. sgRNAs directed at *MyD88*, *IFNAR1*, and *CCR5* were synthesized by Synthego or TriLink Technologies with the three terminal nucleotides in both ends chemically modified with 2′-O-methyl-3′-phosphorothioate[35]. Thawed CD34$^+$ HSPC were initially pre-cultured at low density ($10^5$ cells/mL) for 3 days in CD34$^+$ HSPC medium (SFEM II medium (STEMCELL Technologies) supplemented with 20 units/mL penicillin, 20 mg/mL streptomycin, Flt3-L (100 ng/mL), SCF (100 ng/mL), TPO (100 ng/mL), IL-6 (100 ng/mL), SR1 (0.75 μM), and UM171 (35 nM)) before being nucleoporated as shown previously[37–39]. Ribonucleoprotein (RNP) complexes were made by incubating Cas9 protein (Integrated DNA Technologies) with sgRNA at a molar ratio of 1:2.5 at 25 °C for 10 min prior to nucleoporation. Nucleoporation was performed using the Lonza 4D-Nucleofector$^{TM}$ System (program DZ100). After a 3-day recovery phase in CD34$^+$ HSPC expansion medium, HSPC medium was changed to the medium promoting pDC differentiation as previously described.

**Statistical analysis**. All data were plotted using GraphPad Prism 6.0 (GraphPad Software, San Diego, CA, USA). Data presented are expressed as means ± standard error of mean (±SEM). Statistical analysis was performed using one-way ANOVA or two-way ANOVA, followed by Bonferroni post hoc test, as indicated in the figure legends. $*P \leq 0.05$, $**P \leq 0.01$, $***P \leq 0.001$, $****P \leq 0.0001$.

## Data availability
All relevant datasets generated during and/or analyzed during the current study are available from the corresponding author upon request.

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

## Acknowledgements

This work was primarily supported by grants from the Danish Council for Independent Research (4004-00237), The Lundbeck Foundation (R194-2015-1388), A.P. Møller foundation, NIH (AI127219), and Frode V. Nyegaard og Hustrus Fond. We would like to thank the FACS Core facility at Aarhus University for providing invaluable help with flow cytometry. A special thanks to Ane Kjeldsen for technical assistance and to Karen Marie Juul Sørensen for designing all graphical illustrations used within the figures.

## Author contributions

A.L., R.O.B., N.R.R., M.P., M.N., P.W.D., and M.R.J. designed the experiments. A.L., C.K., R.O.B., L.K., J.H.E., C.C.P., S.P., H.Q.T., N.U., N.R.R., and M.N. performed the experiments and analysis. A.L., R.O.B., N.R.R., M.N., P.W.D., and M.R.J. wrote the paper and all authors contributed to the manuscript writing.
