## [Peer Review File · Nature Communications]

Reviewers' comments:

Reviewer #1 (Remarks to the Author):

pDC are important type I IFN producers after stimulation with many different viruses. In the human system, so far studies towards the biology of pDC were limited by the little cell yield either of preparation methods from blood, or of in vitro differentiation protocols, as well as by donor variations. Furthermore, for the same reasons genetic modification of primary human pDC was basically impossible. In this situation, Laustsen and colleagues invested in the systematic analysis and improvement of in vitro differentiation protocols of pDCs from hematopoietic stem and progenitor cells (HSPC). Finally, they were able to establish an overall simple and rather efficient pDC differentiation protocol. In a first step pDC precursors are differentiated from CD34+ HSPC, which in a second step are primed in the presence of low level IFN-beta and IFN-gamma to yield pDC that resemble in many aspects those in the blood stream. Furthermore, the authors verified the suitability of their approach also for genetic manipulation by applying CRISPR/Cas9 mediated gene editing for the adapter MyD88 and the cytokine receptor IFNAR1. Experiments with IFNAR1 deficient pDC indicated the key relevance of the type I IFN axis for the induction of type I IFN responses upon pDC stimulation with TLR7 or 9 agonists. This is a very interesting study with carefully planned experiments and high quality data. In particular the in vitro differentiation protocol looks very convincing and is well documented by numerous statistically relevant data. Collectively, this study can be considered as a true break-through in the field of human pDC biology which is of interest of a broad readership.

Nevertheless, although the authors show convincing phenotypic and functional data of their HSPC-derived pDCs compared with blood-derived pDC, a more thorough RNAseq analysis of in vitro differentiated precursor pDC, primed pDC and stimulated pDC compared with blood derived naïve and stimulated pDC would help to further highlight the relevance of the in vitro differentiation protocol. This is an important experiment to be included in order to make this manuscript a true hall mark publication defining the state-of-the-art in human pDC biology for the next years.

The authors refer to the antigen presenting cell function of pDC and determine the expression of CD40, CD80, CD83, CD86 on in vitro differentiated pDC precursors as well as on IFN-beta and gamma primed pDC. However, the tested markers alone do not report on the antigen presenting cell function, yet. Therefore, it is recommended to include experiments with protein loaded pDC that restimulate expansion of antigen-specific T cells. Such experiments can be easily conducted e.g. with CMV-specific T cells that in some healthy donors are present with high abundance. Alternatively, the authors should use a slightly more moderate formulation not putting the antigen presenting cell characteristics in the centre.

Reviewer #2 (Remarks to the Author):

This manuscript claims to have identified a new human pDC precursor that arises following culture of human CD34 progenitors in vitro. In vitro culture methods for generating human pDC have been previously described. Here the authors use a new combination of factors, all of which have previously been shown to be involved in pDC development/differentiation, to refine a culture system that yields significantly larger numbers of pDC than those previously published. Compared to blood pDC, the in vitro derived pDC express lower levels of known pDC phenotypic markers and are poorly responsive to TLR7 and TLR9 stimulation, which can be improved following stimulation with IFN, leading the authors to conclude that these are a new precursor pDC population. Although the identification of a precursor pDC population would certainly be of interest to the field, there are significant issues that require addressing before such conclusions can be made.

Major points:

The pDC were enriched from the culture by magnetically selecting lin-CD11c- cells. Further

characterisation of this population is required in order to demonstrate that these are bonafide pDC or specific pDC precursors. Of particular concern are the pre-cDC, defined as lin-DR+ CD123+CD33+CD45RA+, that give rise to cDC1 and cDC2 but not pDC. These also express classical pDC markers including CD123, CD303 and CD304 (See et al Science 356 2017) and could therefore be contaminating the population of interest the authors refer to as HSC-pDC. In order to demonstrate that the HSC-pDC population are genuine pDC precursors the following should be performed:

1. More extensive phenotyping on the HSC-pDC population, including markers such CD33, CX3CR1, CD2, CD5 and CD327 recently described to segregate pDC from pre-cDC (See et al Science 356 2017). CD34 and CD45RA should also be included in order to compare with a previously reported pDC precursor defined as CD34+CD45RA+CD4+CD123+ (Blom et al J Exp Med 2000).
2. Stimulation of HSC-pDC with cDC activating stimuli (eg poly I:C, LPS and or R848) and analysis of cDC cytokines such as IL-12.
3. Further culture of HSC-pDC in cDC conditioning media and subsequent analysis of cDC subset phenotype and function in order to demonstrate that these are precursors of pDC and not other DC subsets.
4. Transcriptome analysis of HSC-pDC compared to pDC, cDC and pre-cDC.
5. Identification of cells with the pDC precursor phenotype in human cord, peripheral blood and or bone marrow.

Other points:

1. Line 279-283 and Fig 2g- the conclusion from the intracellular DNA experiment is that the HSC-pDC have capacity to induce innate signals. However, like the TLR stimulation these appear to be significantly lower than blood DC.
2. Additional replicates should be included for knockdown of IFNAR (Fig 5)– only 2 data points are shown.
3. Line 397, Discussion first sentence statement that IFN priming is essential for the ability of pDC to mount a robust and efficient innate immune response is substantiated by the data as presented.
4. Line 607 “linage” is a typo

Reviewer #3 (Remarks to the Author):

In the manuscript, Laustsen et al. describe a novel protocol for generating functional human plasmacytoid dendritic cells (pDCs) from CD34+ umbilical cord precursor cells. They also present a protocol for modifying these in vitro-derived pDCs using CRISPR/Cas9, and show evidence for the involvement of IFNAR1 and MyD88 in the functional maturation of pDCs using their protocol.

The study is very well conducted, and as the function of pDCs is understudied the presented protocols and findings are of great importance for future discovery.

The method section is well described, besides information about the serum free-culture system that was used (mentioned on line 313).

Regarding the statistics, I lack a bit of clarification. It should be defined what they refer to with *, **, and ***, especially since they use one-way (and two-way) ANOVAs as well as Bonferroni correction which affects the definitions of what is significant. In general, I am a bit skeptical of using one-way ANOVAs and I think it should be specified in the figure legend which test was used in the individual experiments.

Major points

The authors convincingly describe an immature precursor pDC that needs a mix of type 1 and 2

IFNs for maturation. This is interesting, but unless this precursor could be found in primary human samples, these cells could very well just be a consequence of the specific culturing protocol they use, and not represent a unique precursor cell. Can the authors show evidence for these precursors in vivo? If not, I think the novel precursor cell conclusion should be toned down.

Minor points

The paper is in general very well written. However, the abstract would benefit from some minor changes to be more clear. I am specifically thinking of this sentence: "These precursor pDCs are found to have low surface expression of pDC markers and reduced type I and IL-6 responses upon stimulation with synthetic TLR7 and TLR9 agonists, but not intracellular sensors". With "reduced type I", they should write out that they are referring to "reduced type 1 interferons". The last part of the sentence ("but not intracellular sensors") should also be clarified; I guess they could write something like "but not with agonists for intracellular DNA/RNA sensors" to make it more clear for an audience that lacks the specific knowledge of the field.

IFN γ and IFN β are very different from many/most points of view (cells producing the cytokines, type of infection triggering the response, signaling pathways, etc). Could any statement about the mixed use of IFN γ and IFN β used in the stimulation cocktail be made? Based on the supplemental data, it looks like the IFN γ has most of the activity, still IFNAR1 is needed. Does IFN γ trigger a type 1 IFN response that could bypass the need for adding IFN β , but still need the IFNAR1?

On line 82-83 it is mentioned that 2'-O-methyl modifications can help to reduce the innate immune response as genetic material is introduced into a cell. I think this is a very important point talking about introducing foreign genetic material into immune cells in general and even more talking about pDCs. I think the manuscript would benefit from highlighting this a bit more, as this would be my main concern related to doing assays with siRNA or CRISPR modified pDCs. I find several references in this review (PMID: 22432611) that for example could be included to support this point.

On line 313 it is mentioned that a serum-free medium protocol was established. I cannot find any details about this. I suggest to describe the protocol and some data or to remove this part. I think it would be good to include it.

On Line 344 it is stated that all blood pDCs are dead at day 8. Looking at the data (fig 4b, and sup fig 12b) this is not true. I suggest writing that <10% are alive at day 8 instead of saying that all are dead.

The efficiency of the CRISPR induced indels is very impressive. Still, the frequency is (as expected) not 100%. Based on fig 5c a/c, it should be ~20% of the cells in the culture that are WT for MyD88. Would you not expect these cells to be able to produce Type 1 IFNs? For the TLR9 stimulation, it looks like the production is completely blocked. How can this be explained, is there an amplification loop involved that needs a certain amount of WT cells to be functional?

A blot showing the efficiency of the IFNAR1 knockdown, similar to fig 5e, should be presented.

Regarding figure 5h and 5i. Could you also present IL-6 data from these conditions?

Response to Reviewers

We thank the reviewers for their positive and constructive comments.

We are pleased to learn that all reviewers find the work of potential significance. We agree on most of the comments and are pleased to now submit a revised manuscript, where we have addressed the majority of the points raised by the reviewers. In particular, we have included data demonstrating that our population of pDCs are not contaminated by cDCs; we show that pDCs can take up and present CMV antigens; and transcriptomic analysis illustrate comparable biological pathways activated by TLR7 stimulation from both blood pDCs and HSPC-pDCs. Also, we have rewritten multiple sections accordingly to the reviewers comments. All in all, we find that the new data strengthen and support the main conclusions of the original manuscript.

Please find a point-by-point reply specifying how we have dealt with the comments of the reviewers. We hope that the revised draft is now acceptable for publication in *Nature Communication*

Reviewer #1 (Remarks to the Author):

pDC are important type I IFN producers after stimulation with many different viruses. In the human system, so far studies towards the biology of pDC were limited by the little cell yield either of preparation methods from blood, or of in vitro differentiation protocols, as well as by donor variations. Furthermore, for the same reasons genetic modification of primary human pDC was basically impossible. In this situation, Laustsen and colleagues invested in the systematic analysis and improvement of in vitro differentiation protocols of pDCs from hematopoietic stem and progenitor cells (HSPC). Finally, they were able to establish an overall simple and rather efficient pDC differentiation protocol. In a first step pDC precursors are differentiated from CD34+ HSPC, which in a second step are primed in the presence of low level IFN-beta and IFN-gamma to yield pDC that resemble in many aspects those in the blood stream. Furthermore, the authors verified the suitability of their approach also for genetic manipulation by applying CRISPAR/Cas9 mediated gene editing for the adapter MyD88 and the cytokine receptor IFNAR1. Experiments with IFNAR1 deficient pDC indicated the key relevance of the type I IFN axis for the induction of type I IFN responses upon pDC stimulation with TLR7 or 9 agonists. This is a very interesting study with carefully planned experiments and high quality data. In particular the in vitro differentiation protocol looks very convincing and is well documented by numerous statistically relevant data. Collectively, this study can be considered as a true break-through in the field of human pDC biology which is of interest of a broad readership.

Major points

1. Nevertheless, although the authors show convincing phenotypic and functional data of their HSPC-derived pDCs compared with blood-derived pDC, a more thorough RNAseq analysis of in vitro differentiated precursor pDC, primed pDC and stimulated pDC compared with blood derived naïve and stimulated pDC would help to further highlight the relevance of the in vitro differentiation protocol. This is an important experiment to be included in order to make this manuscript a true hall mark publication defining the state-of-the-art in human pDC biology for the next years.

We agree with the reviewer that transcriptomic analysis can support the functional and phenotypic data on our pDC model. We have now conducted an RNAseq experiment as suggested by the reviewer, and included the bioinformatics analysis of these datasets in figure 4. Our findings illustrates that treatment of blood pDCs with TLR7 agonists activate

similar gene pathways as that observed when primed HSPC-pDCs are activated with TLR7 agonists.

2. The authors refer to the antigen presenting cell function of pDC and determine the expression of CD40, CD80, CD83, CD86 on in vitro differentiated pDC precursors as well as on IFN- β and gamma primed pDC. However, the tested markers alone do not report on the antigen presenting cell function, yet. Therefore, it is recommended to include experiments with protein loaded pDC that restimulate expansion of antigen-specific T cells. Such experiments can be easily conducted e.g. with CMV-specific T cells that in some healthy donors are present with high abundance. Alternatively, the authors should use a slightly more moderate formulation not putting the antigen presenting cell characteristics in the centre.

We thank the reviewer for these suggestions. As illustrated in the new figure 5b-d and line 448-4454, we now demonstrate that primed pDCs are capable of presenting CMV antigens, thereby enabling a robust IFN-gamma response from CMV+ donor PBMCs.

Reviewer #2 (Remarks to the Author):

This manuscript claims to have identified a new human pDC precursor that arises following culture of human CD34 progenitors in vitro. In vitro culture methods for generating human pDC have been previously described. Here the authors use a new combination of factors, all of which have previously been shown to be involved in pDC development/differentiation, to refine a culture system that yields significantly larger numbers of pDC than those previously published. Compared to blood pDC, the in vitro derived pDC express lower levels of known pDC phenotypic markers and are poorly responsive to TLR7 and TLR9 stimulation, which can be improved following stimulation with IFN, leading the authors to conclude that these are a new precursor pDC population. Although the identification of a precursor pDC population would certainly be of interest to the field, there are significant issues that require addressing before such conclusions can be made.

Major points: The pDC were enriched from the culture by magnetically selecting lin-CD11c- cells. Further characterisation of this population is required in order to demonstrate that these are bonafide pDC or specific pDC precursors. Of particular concern are the pre-cDC, defined as lin-DR+ CD123+CD33+CD45RA+, that give rise to cDC1 and cDC2 but not pDC. These also express classical pDC markers including CD123, CD303 and CD304 (See et al Science 356 2017) and could therefore be contaminating the population of interest the authors refer to as HSC-pDC. In order to demonstrate that the HSC-pDC population are genuine pDC precursors the following should be performed:

1. More extensive phenotyping on the HSC-pDC population, including markers such CD33, CX3CR1, CD2, CD5 and CD327 recently described to segregate pDC from pre-cDC (See et al Science 356 2017).
CD34 and CD45RA should also be included in order to compare with a previously reported pDC precursor defined as CD34+CD45RA+CD4+CD123+ (Blom et al J Exp Med 2000).
2. Stimulation of HSC-pDC with cDC activating stimuli (eg poly I:C, LPS and or R848) and analysis of cDC cytokines such as IL-12.
3. Further culture of HSC-pDC in cDC conditioning media and subsequent analysis of cDC subset phenotype and function in order to demonstrate that these are precursors of pDC and not other DC subsets.
4. Transcriptome analysis of HSC-pDC compared to pDC, cDC and pre-cDC.

5. Identification of cells with the pDC precursor phenotype in human cord, peripheral blood and or bone marrow.

We agree with the reviewer that magnetically selecting the pDCs could yield contaminants and that additional controls of purity and function would benefit our study. We have included new supplementary figures 9-11 and a new figure 4, described in line 376-405, 416-438, and 561-5567 of the manuscript, to address the above five points. In summary our data support the finding that HSPC-pDCs: 1) Do not express pre-DC markers prior to or after IFN priming; 2) Do not produce IL12; 3) Have pathways upregulated after TLR7 treatment that parallel that of TLR7 stimulation of blood pDCs, as determined by RNA-seq. With regards to identification of cells with the pDC precursor phenotype, we focused on cord blood (CB) as there are substantially more pDCs in CB than in peripheral blood, and cells from the human bone marrow are generally difficult to obtain. Our flow-based analysis of CB-derived pDCs does indeed identify a CD123^{dim}CD303^{dim} population similar to that observed for our day 21 HSPC-pDCs, which upon IFN priming becomes CD123^{high}CD303^{high}. These data are described in supplementary figure 9, 10, 11,

Other points:

1. Line 279-283 and Fig 2g- the conclusion from the intracellular DNA experiment is that the HSC-pDC have capacity to induce innate signals. However, like the TLR stimulation these appear to be significantly lower than blood DC.

We agree that the IFN signal is lower comparing blood pDCs with HSPC-pDCs. However, the point we wanted to make is that IFN induction is not absent as observed for TLR agonists. To clarify, we have rephrased the sentence in line 341-343

2. Additional replicates should be included for knockdown of IFNAR (Fig 5)– only 2 data points are shown.

We have now repeated these CRISPR/Cas9 experiments on three additional donors. These data are presented in figure 6 and support our original findings and conclusions.

3. Line 397, Discussion first sentence statement that IFN priming is essential for the ability of PDC to mount a robust and efficient innate immune response is substantiated by the data as presented.

We find that our data support such statement

4. Line 607 “linage” is a typo

We have now corrected this typo.

Reviewer #3 (Remarks to the Author):

In the manuscript, Laustsen et al. describe a novel protocol for generating functional human plasmacytoid dendritic cells (pDCs) from CD34+ umbilical cord precursor cells. They also present a protocol for modifying these in vitro-derived pDCs using CRISPR/Cas9, and show evidence for the involvement of IFNAR1 and MyD88 in the functional maturation of pDCs using their protocol. The study is very well conducted, and as the function of pDCs is understudied the presented protocols and findings are of great importance for future discovery.

Points raised:

1. The method section is well described, besides information about the serum free-culture system that was used (mentioned on line 313).

We have now added more information about the serum free-conditions, in supplementary figure 8, and in manuscript lines 372-374 and in method section lines 127-129.

2. Regarding the statistics, I lack a bit of clarification. It should be defined what they refer to with *, ** and ***, especially since they use one-way (and two-way) ANOVAs as well as Bonferroni correction which affects the definitions of what is significant. In general, I am a bit skeptical of using one-way ANOVAS and I think it should be specified in the figure legend which test was used in the individual experiments.

We appreciate the comment by the reviewer. All figure legends do now include information about the statistical analysis used, and we have confirmed the usability of the tests with a biostatistician.

3. The authors convincingly describe an immature precursor pDC that needs a mix of type 1 and 2 IFNs for maturation. This is interesting, but unless this precursor could be found in primary human samples, these cells could very well just be a consequence of the specific culturing protocol they use, and not represent a unique precursor cell. Can the authors show evidence for these precursors in vivo? If not, I think the novel precursor cell conclusion should be toned down.

We thank the reviewer for the comment, which was also posed by reviewer #2. As described above (point 5 under reviewer 2), we have been able to identify a cell population in cord blood with striking similarity to our HSPC-pDCs before and after IFN priming. These data are presented in supplementary figure 11 and manuscript line 395-405 and 561-567

Minor points

4. The paper is in general very well written. However, the abstract would benefit from some minor changes to be more clear. I am specifically thinking of this sentence: “These precursor pDCs are found to have low surface expression of pDC markers and reduced type I and IL-6 responses upon stimulation with synthetic TLR7 and TLR9 agonists, but not intracellular sensors”. With “reduced type I”, they should write out that they are referring to “reduced type I interferons”. The last part of the sentence (“but not intracellular sensors”) should also be clarified; I guess they could write something like “but not with agonists for intracellular DNA/RNA sensors” to make it more clear for an audience that lacks the specific knowledge of the field.

We have now altered the abstract to make these points clearer.

5. IFN γ and IFN β are very different from many/most points of view (cells producing the cytokines, type of infection triggering the response, signaling pathways, etc). Could any statement about the mixed use of IFN γ and IFN β used in the stimulation cocktail be made? Based on the supplemental data, it looks like the IFN γ has most of the activity, still IFNAR1 is needed. Does IFN γ trigger a type 1 IFN response that could bypass the need for adding IFN β , but still need the IFNAR1?

We thank the reviewer for these very interesting points. We have added a section in the discussion (lines 578-590) addressing these these points.

6. On line 82-83 it is mention that 2'-O-methyl modifications can help to reduce the innate immune response as genetic material is introduced into a cell. I think this is a very important point talking about introducing foreign genetic material into immune cells in general and even more talking about pDCs. I think the manuscript would benefit from highlighting this a bit more, as this would be my main concern related to doing assays with siRNA or CRISPR modified pDCs. I find several references in this review (PMID: 22432611) that for example could be included to support this point.

We thank the reviewer for this suggestion, and have highlighted this in the introduction (lines 86-94) and in the discussion.

7. On line 313 it is mention that a serum-free medium protocol was established. I cannot find any details about this. I suggest to describe the protocol and some data or to remove this part. I think it would be good to include it.

We have now included these data in the methods section (lines 125-128).

8. On Line 344 it is stated that all blood pDCs are dead at day 8. Looking at the data (fig 4b, and sup fig 12b) this is not true. I suggest writing that <10% are alive at day 8 instead of saying that all are dead.

We agree with the reviewer and have modified the text accordingly.

9. The efficiency of the CRISPR induced indels is very impressive. Still, the frequency is (as expected) not 100%. Based on fig 5c a/c, it should be ~20% of the cells in the culture that are WT for MyD88. Would you not expect these cells to be able to produce Type 1 IFNs? For the TLR9 stimulation, it looks like the production is completely blocked. How can this be explained, is there an amplification loop involved that needs a certain amount of WT cells to be functional?

We thank the reviewer for this comment. We note that the TIDE assay used to detect Indels sometimes underestimate Indels frequencies due to sensitivity issues caused by quality of the Sanger Sequencing. Therefore, for some samples Indel frequencies may be even higher than reported and we note that we have detected Indel frequencies of up to 97% at IFNAR1 and CCR5. We have performed the same experiments for an additional three donors and observe some levels of type I IFN upon 80% MyD88 knockdown, which likely reflect activity of some remaining MyD88. However, we also agree with the reviewer that an amplification loop could be in play, and have indicated this in the manuscript (lines 511-517)

10. A blot showing the efficiency of the IFNAR1 knockdown, similar to fig 5e, should be presented.

We agree this would be a nice control to show. However, we have not been able to identify good antibodies that can be used to confirm protein knockdown of IFNAR1 receptor. However, we have added a western blot demonstrating that STAT1 is not phosphorylated after IFN- β stimulation, illustrating that the IFNAR receptor is either absent or at least unable to signal (Figure. 5e). As a control, we have confirmed that signaling through the type II IFN receptor is intact as shown in Suppl. Fig. 16e.

11. Regarding figure 5h and 5i. Could you also present IL-6 data from these conditions?

We have now included a new figure 5i that illustrates the levels of IL-6 in sgRNA-targeted HSPC-pDCs for three new donors.

REVIEWERS' COMMENTS:

Reviewer #1 (Remarks to the Author):

Thank you to the authors for the detailed response to the reviewers' comments and the additional experimental work they performed. The new data further substantiate the conclusions drawn by the authors.

However, one semantic inaccuracy still exists in the manuscript: In line 456 and 457 the authors wrote "to extend these observations, we next investigated if HSPC-pDCs were capable of presenting antigens and activating T cells. For this, we loaded IFN-primed and TLR7 activated HSPC-pDCs with peptides derived from CMV...". Although it is appreciated that the authors experimentally studied whether peptide loaded HSPC-pDC were able to restimulate antigen-specific T cells, this assay certainly does not address antigen presentation. For this the authors would have had to load HSPC-pDCs with a recombinant CMV protein such as the pp65 antigen in order to then study T cell restimulation. Since this experiment has not been performed it is recommended that the opening sentence is toned down a bit. In line 460 the authors should also complete their statement "... pre-loaded with CMV peptide" because the HSPC-pDCs were not loaded with CMV. It is appreciated that in the end the authors correctly concluded that "... together, these data indicate that activated HSPC-pDCs can present peptides and activate antigen-specific T cells".

Reviewer #2 (Remarks to the Author):

The revisions by Lausten et al are significant. A substantial body of new data has been added and discussed that significantly enhances the quality and conclusions of the manuscript. The authors have adequately addressed the concerns raised.

Reviewer #3 (Remarks to the Author):

The authors have answered my questions well and adjusted the manuscript accordingly. I do like the study a lot.

Although it's beyond my original questions, I feel that I need to address the response to question 2 from reviewer 1. I do think the experiment 5b-d lack some obvious controls. It's not clear that the pDCs are the cells actually presenting the CMV peptide, as the PBMC contains other potential APC that could access the peptides. The washing of the pDCs after pulsing with the CMV peptides obviously should get rid of most peptide that is not taken up by the pDCs, but likely some are still left or could be released from the pDCs to other cells during the 24h co-culture.

Also, it's not clear that the IFN γ measured by ELISA from the supernatant is coming from T cells. In PBMC there are also other potent IFN γ producers, like NK cells, that would be involved in the immune response against CMV. Likely NK cells wouldn't be activated by CMV peptides, but it's not impossible that the used peptide-mix contains something that directly or indirectly would activate e.g. NK cells.

To me, it seems like using purified T cells from CMV+ patients and incubating these with the pDCs +/- peptide-pulsed, measuring IFN γ production, could be an easy way to prove both my concerns above wrong. If the CMV peptide mix is also added to the purified T cells alone, and no IFN γ is produced, the experimental setup would be further controlled for the presence of other APCs with the ability to activate the IFN γ response.

Furthermore, I don't find information that shows what differs between fig 5c and 5d.

Beyond this comment, I'm very satisfied with the response to the review questions and think the manuscript is clearly suitable for publication.

Response to Reviewers

We thank the reviewers for their positive and constructive comments.

We are pleased to learn that all reviewers acknowledge that the new set of data further validates our conclusions and make it appropriate for publication. Please find a point-by-point reply specifying how we have dealt with the final comments raised by the reviewers.

Reviewer #1 (Remarks to the Author):

Thank you to the authors for the detailed response to the reviewers' comments and the additional experimental work they performed. The new data further substantiate the conclusions drawn by the authors.

However, one semantic inaccuracy still exists in the manuscript: In line 456 and 457 the authors wrote "to extend these observations, we next investigated if HSPC-pDCs were capable of presenting antigens and activating T cells. For this, we loaded IFN-primed and TLR7 activated HSPC-pDCs with peptides derived from CMV...". Although it is appreciated that the authors experimentally studied whether peptide loaded HSPC-pDC were able to restimulate antigen-specific T cells, this assay certainly does not address antigen presentation. For this the authors would have had to load HSPC-pDCs with a recombinant CMV protein such as the pp65 antigen in order to then study T cell restimulation. Since this experiment has not been performed it is recommended that the opening sentence is toned down a bit. In line 460 the authors should also complete their statement "... pre-loaded with CMV peptide" because the HSPC-pDCs were not loaded with CMV. It is appreciated that in the end the authors correctly concluded that "... together, these data indicate that activated HSPC-pDCs can present peptides and activate antigen-specific T cells".

It is true that we did not use a single pp65 antigen to pulse the HSPC-pDCs. However, we used a broader antigen mix composed of more than 14 different CMV peptides, defined as HLA class I-restricted T cell epitopes. This mixture did include 8 different HCMV pp65 antigens. We used this mixed peptide cocktail to pulse HSPC-pDCs for 3 hrs, substantial washing and then combined them with PBMCs from patients with known CMV seroconversion.

We agree with the reviewer that we have made some semantic errors. These are now corrected and we have improved the clarification of the sentence in the manuscript.

Reviewer #2 (Remarks to the Author):

The revisions by Lausten et al are significant. A substantial body of new data has been added and discussed that significantly enhances the quality and conclusions of the manuscript. The authors have adequately addressed the concerns raised.

We thank the reviewer for the positive feedback

Reviewer #3 (Remarks to the Author):

The authors have answered my questions well and adjusted the manuscript accordingly. I do like the study a lot.

Although it's beyond my original questions, I feel that I need to address the response to question 2 from reviewer 1. I do think the experiment 5b-d lack some obvious controls. It's not clear that the pDCs are the cells actually presenting the CMV peptide, as the PBMC contains other potential APC that could access the peptides. The washing of the pDCs after pulsing with the CMV peptides obviously should get rid of most peptide that is not taken up by the pDCs, but likely some are still left or could be released from the pDCs to other cells during the 24h co-culture.

The PBMC were first cultured for themselves prior to mixture with the CMV-peptide pulsed HSPC-pDCs, thus we got rid of monocytes, but we acknowledge that small amounts of other APCs could still be present.

Also, it's not clear that the IFN γ measured by ELISA from the supernatant is coming from T cells. In PBMC there are also other potent IFN γ producers, like NK cells, that would be involved in the immune response against CMV. Likely NK cells wouldn't be activated by CMV peptides, but it's not impossible that the used peptide-mix contains something that directly or indirectly would activate e.g. NK cells.

We agree with the reviewer that we cannot exclude this phenomenon 100%

To me, it seems like using purified T cells from CMV+ patients and incubating these with the pDCs +/- peptide-pulsed, measuring IFN γ production, could be an easy way to prove both my concerns above wrong. If the CMV peptide mix is also added to the purified T cells alone, and no IFN γ is produced, the experimental setup would be further controlled for the presence of other APCs with the ability to activate the IFN γ response.

We agree with the reviewer that this could have been a supportive experiment to the data presented in figure 5.

Furthermore, I don't find information that shows what differs between fig 5c and 5d.

We agree that the sentence may have been slightly confusing and thus have altered it to describe in details the differences between figure 5c and d.

Beyond this comment, I'm very satisfied with the response to the review questions and think the manuscript is clearly suitable for publication.